# The *Drosophila* formin Fhos is a primary mediator of sarcomeric thin-filament array assembly

**Arkadi Shwartz, Nagaraju Dhanyasi, Eyal D Schejter\*, Ben-Zion Shilo\***

Department of Molecular Genetics, Weizmann Institute of Science, Rehovot, Israel

**Abstract** Actin-based thin filament arrays constitute a fundamental core component of muscle sarcomeres. We have used formation of the *Drosophila* indirect flight musculature for studying the assembly and maturation of thin-filament arrays in a skeletal muscle model system. Employing GFP-tagged actin monomer incorporation, we identify several distinct phases in the dynamic construction of thin-filament arrays. This sequence includes assembly of nascent arrays after an initial period of intensive microfilament synthesis, followed by array elongation, primarily from filament pointed-ends, radial growth of the arrays via recruitment of peripheral filaments and continuous barbed-end turnover. Using genetic approaches we have identified Fhos, the single *Drosophila* homolog of the FHOD sub-family of formins, as a primary and versatile mediator of IFM thin-filament organization. Localization of Fhos to the barbed-ends of the arrays, achieved via a novel N-terminal domain, appears to be a critical aspect of its sarcomeric roles.

## Introduction

Sarcomeres constitute the basic functional units of muscle fibers, endowing these large and specialized cells with their contractile capacity. Central to sarcomere function is the lattice-like organization of two filament systems: an actin-based thin-filament array, which provides a stiff backbone along which thick filaments, composed of myosin motor proteins, 'slide' in order to produce force and contractile motion (*Squire, 1997*). The spatial organization and efficient operation of this remarkable cellular machinery relies on a host of dedicated proteins and protein complexes, which act to regulate sarcomere size and streamline its activity, and to coordinate between the multiple sarcomeric units that comprise individual myofibrils (*Clark et al., 2002*; *Ehler and Gautel, 2008*; *Gautel and Djinovic-Carugo, 2016*).

Despite their fundamental significance, elucidation of the molecular mechanisms underlying assembly, maturation and maintenance of thin-filament arrays remains one of the major open issues in the study of sarcomere structure and function. While mechanisms relating to size definition and stability of the arrays have been extensively investigated (*Fernandes and Schock, 2014*; *Meyer and Wright, 2013*), other key aspects of microfilament array formation and dynamics, including determination of distinct phases of array maturation, the identity and regulation of elements mediating filament nucleation/elongation, and the processes governing incorporation of additional filaments into nascent arrays are not resolved (*Ono, 2010*).

Here we address these matters in the context of formation and development of the *Drosophila* indirect flight muscles (IFMs). These are the largest muscles of the adult fly, which power flight by regulated contraction of the thorax (*Dickinson, 2006*). A major subset of the IFMs, the dorso-longitudinal muscles (DLMs), closely resemble vertebrate skeletal muscles in both their developmental program and in their mature myofibrillar structure (*Dutta and VijayRaghavan, 2006*; *Roy and VijayRaghavan, 1999*), making them a particularly attractive model system, in which the powerful

**\*For correspondence:** eyal.
schejter@weizmann.ac.il (EDS);
benny.shilo@weizmann.ac.il (B-ZS)

**Competing interests:** The authors declare that no competing interests exist.

**eLife digest** Muscles owe their ability to contract to structural units called sarcomeres, and a single muscle fiber can contain many thousands of these structures, aligned one next to the other. Each mature sarcomere is made up of precisely arranged and intertwined thin filaments of actin and thicker bundles of motor proteins, surrounded by other proteins. Sliding the motors along the filaments provides the force needed to contract the muscle. However, it was far from clear how sarcomeres, especially the arrays of thin-filaments, are assembled from scratch in developing muscles.

When the fruit fly *Drosophila* transforms from a larva into an adult, it needs to build muscles to move its newly forming wings. While smaller in size, these flight muscles closely resemble the skeletal muscles of animals with backbones, and therefore serve as a good model for muscle formation in general. New muscles require new sarcomeres too, and now Shwartz et al. have observed and monitored sarcomeres assembling in developing flight muscles of fruit flies, a process that takes about three days.

The analysis made use of genetically engineered flies in which the gene for a fluorescently labeled version of actin, the building block of the thin filaments, could be switched on at specific points in time. Looking at how these green-glowing proteins become incorporated into the growing sarcomere revealed that the assembly process involves four different phases. First, a large store of unorganized and newly-made thin filaments is generated for future use. These filaments are then assembled into rudimentary structures in which the filaments are roughly aligned. Once these core structures are formed, the existing filaments are elongated, while additional filaments are brought in to expand the structure further. Finally, actin proteins are continuously added and removed at the part of the sarcomere where the thin filaments are anchored.

Shwartz et al. went on to identify a protein termed Fhos as the chief player in the process. Fhos is a member of a family of proteins that are known to elongate and organize actin filaments in many different settings. Without Fhos, the thin-filament arrays cannot properly begin to assemble, and the subsequent steps of growth and expansion are blocked as well.

The next challenges will be to understand what guides the initial stages in the assembly of the thin-filament array, and how the coordination between assembly of actin filament arrays and motor proteins is executed. It will also be important to determine how sarcomeres are maintained throughout the life of the organism when defective actin filaments are replaced, and which proteins are responsible for carrying out this process.

molecular genetic approaches available to study *Drosophila* development can be harnessed to investigate and elucidate general principles of myogenesis.

DLM formation initiates by fusion of hundreds of individual myoblasts to a set of larval muscles during the first 24-30 hours of pupal development (*Fernandes et al., 1991*). The subsequent ~80 hrs of myogenesis leading up to eclosion of the adult fly include formation and maturation of a parallel arrangement of myofibrils and assembly and growth of sarcomeric units within them (*Reedy and Beall, 1993*; *Weitkunat et al., 2014*). This sequence of events takes place over a wide time window, providing an opportunity to temporally dissected and manipulate the coordinated processes giving rise to thin-filament array assembly and maturation.

IFM sarcomeres initiate as small, nascent structures, that grow considerably over the course of pupal development (*Sparrow and Schock, 2009*), reaching a final, uniform size of 3.4 µm in length and 1.5 µm in diameter. Spatial organization of the mature, evenly-spaced IFM sarcomeres closely mirrors that of striated vertebrate skeletal muscle (*Reedy and Beall, 1993*). Individual sarcomeric units are defined by Z-disc borders, which serve as anchoring sites for the barbed ends of the thin-filament arrays, and by a central, microfilament-free H-zone, bordered by the pointed-ends of neighboring thin-filament arrays. We utilized temporally-controlled expression of GFP-tagged actin monomers (*Roper et al., 2005*) to follow thin-filament array dynamics, and recognize key phases and distinct transitions of the arrays throughout sarcomerogenesis. In parallel we identified Fhos, the single *Drosophila* member of the conserved FHOD family of formin proteins (*Schonichen and*

*Geyer, 2010*), as a major contributor to thin-filament array assembly and growth. An important aspect of Fhos involvement is its critical role in radial growth of the arrays, by mediating incorporation of new peripheral filaments to the nascent core structure. The elongation of thin filament arrays, shown to occur primarily from their pointed ends (*Mardahl-Dumesnil and Fowler, 2001*), is mediated by the WH2-domain actin regulator Sals protein. While the combined activities of Fhos and Sals can account for most aspects of thin-filament array growth and maturation during pupal stages, other elements are likely involved in additional processes that shape and maintain the arrays, such as continuous monomer exchange at the barbed-ends.

## Results

### Distinct modes of thin-filament array assembly and growth

Assembly of sarcomeric thin-filament arrays has been traditionally studied by monitoring global changes in sarcomere structure in fixed samples of muscle tissue. However, in order to decipher the underlying regulatory mechanisms and uncover the machineries that drive this process, it is essential to monitor the dynamic patterns of actin monomer incorporation into the growing sarcomere. We therefore followed the incorporation of GFP-tagged actin monomers into IFM sarcomeres, throughout the pupal stages of development, as a means of revealing the assembly and maturation of IFM sarcomeric thin-filament arrays in developing flies. Inducible UAS-based GFP-actin transgenes (*Roper et al., 2005*; *Verkhusha et al., 1999*) have proven to be reliable tools and have been used extensively to study microfilament localization and dynamics during *Drosophila* development (*Fulga et al., 2007*; *Jacinto et al., 2000*; *Kaltschmidt et al., 2002*; *Perkins and Tanentzapf, 2014*; *Schottenfeld-Roames and Ghabrial, 2012*). The actin isoform at chromosomal position 88F, one of six *Drosophila* actin genes, is specifically expressed during the formation of the pupal muscles, and represents the major actin isoform used in this tissue (*Beall et al., 1989*; *Fyrberg et al., 1983*). We therefore employed temporally-restricted induction protocols of UAS-GFP-actin88F (*Roper et al., 2005*), and compared the resulting GFP patterns to the outlines of phalloidin-stained sarcomeres, as a primary tool for following the dynamics of IFM thin-filament array development (*Figure 1A*).

Following continuous expression of GFP-actin88F, a close correspondence between the sarcomeric GFP-actin and phalloidin patterns is observed in IFMs isolated from adult flies (*Figure 1B–B"*). The normal size and appearance of the IFM sarcomeres indicates that the GFP-tagged actin does not block or interfere with the structuring and organization of the thin filament arrays throughout the process, and the overlap between the phalloidin and GFP-actin patterns demonstrates the reliability of this tool for monitoring thin-filament array assembly.

We now attempted to break down into stages the processes of IFM thin-filament array organization and growth, by restricting actin-GFP expression to defined 'time windows' during pupal development. Temporal control of expression was achieved by using the general muscle driver *mef2*-GAL4 (*Ranganayakulu et al., 1996*) in combination with the GAL80^ts/TARGET system (*McGuire et al., 2004*). While the timing of induction of GFP-actin is readily controlled using this system, effective 'chase' is not possible due to the stability of the GFP-actin monomers. We first induced expression of GFP-actin88F at the onset of pupariation, and examined the nascent IFMs at 30 hr APF (after puparium formation) (*Figure 1C–C"*). Individual myofibrils are already apparent at this early stage, displaying a nearly homogeneous distribution of microfilaments and an irregular pattern of nascent Z-disc structures, revealed using the early Z-disc marker Zasp52 (*Jani and Schock, 2007*; *Katzemich et al., 2013*). A full correspondence between the GFP-actin88F and phalloidin patterns is observed, implying that the bulk of early IFM microfilaments are formed by extensive *de novo* filament polymerization.

A very different profile of actin monomer incorporation is observed when a 15 hr pulse of actin88F-GFP expression is provided at 30–45 hr APF, immediately following the initial period of extensive polymerization (*Figure 1D–D"*). The GFP-actin and microfilament distributions become markedly distinct from each other during this phase, in which the patterns of Zasp52, the M-line marker Obscurin (*Burkart et al., 2007*) and phalloidin are now indicative of repeating sarcomeric units (*Figure 1D–D"* and *Figure 1—figure supplement 1B–E*). In contrast to the early 'smeared' pattern that filled the myofibrils, actin-GFP is now restricted to discrete, isolated spots, the great majority of which (~80%) positioned at either or both the ends of the nascent arrays (insets

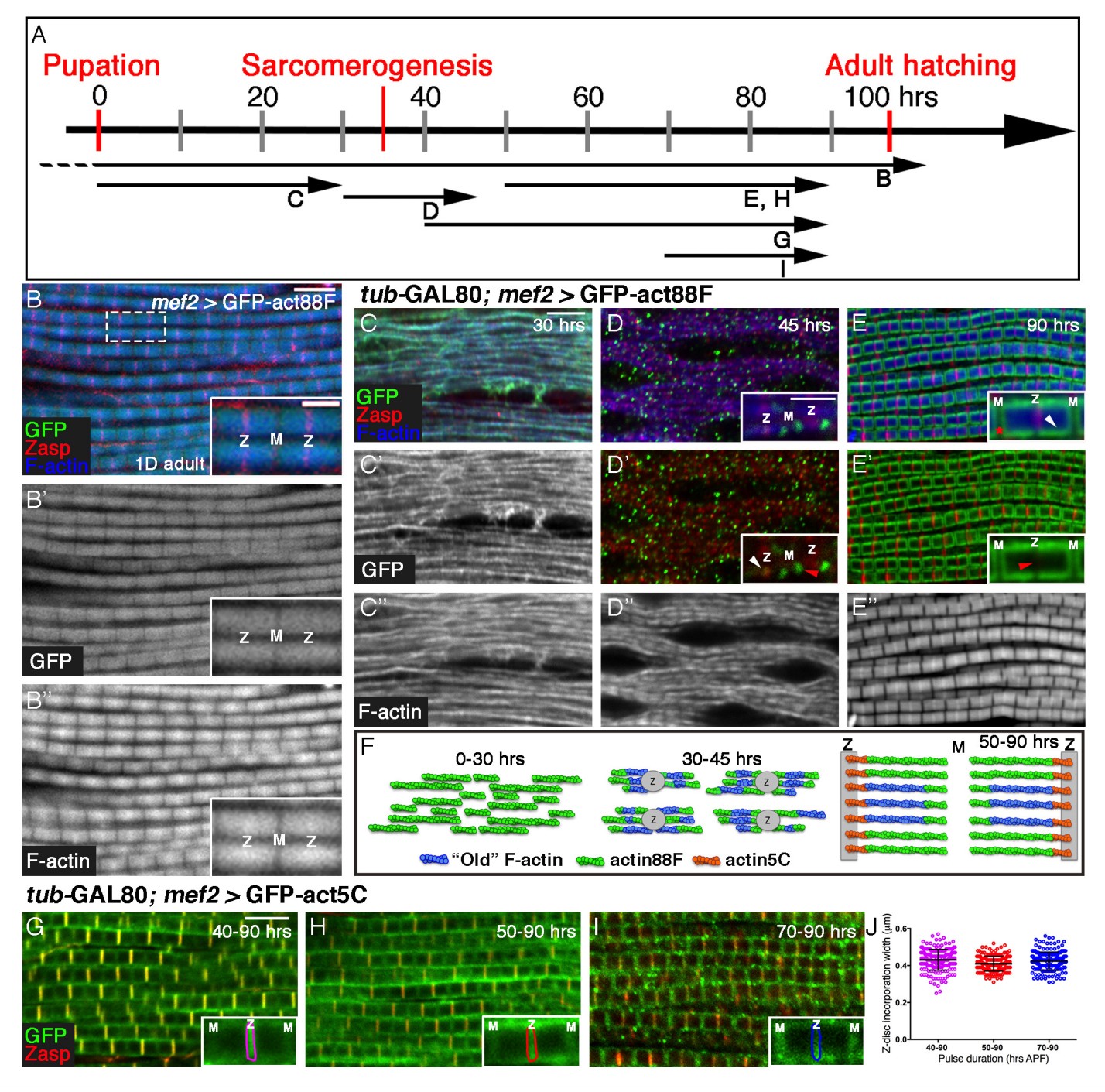

**Figure 1.** Four distinct modes of GFP-actin monomer incorporation contribute to formation of IFM thin-filament arrays. (A) Scheme of IFM development intervals used for unrestricted and temporally restricted expression of GFP-actin88F (B–E") or GFP-*actin5C* (G–I). (B–B") Induction of GFP-actin88F expression (green, gray) with *mef2*-Gal4 throughout fly development results in full monomer incorporation into the thin-filament arrays (phalloidin- blue, gray), as monitored in IFMs of young (1–3 days old) adults. Z-discs are indicated by anti-Zasp52 (red). The designations 'Z' and M' are used throughout to mark the Z-disc (array barbed-end) and H-zone/M line (array pointed end) regions of the sarcomere. (C–E") Incorporation patterns of GFP-actin88F (green, gray) following temporally restricted expression pulses using the *mef2*-GAL4 driver and the GAL80ts/TARGET system. Microfilaments are visualized with phalloidin (blue, gray). Z-discs are indicated by anti-Zasp52 (red). (C–C") 0–30 hrs APF. Initial uniform incorporation. (D–D") 30–45 hrs APF. 'Patched' incorporation of monomers. This mode occurs mainly at array ends (insets in D and D'), in proximity to the future Z-disc (Z, white arrow) or towards the opposite boundary of the nascent arrays (M, red arrow). (E–E") 50–90 hr APF. Monomer incorporation into a 'frame' generated by peripheral 'thickening' (white arrowhead in panel E inset) and pointed-end growth (M and red asterisk in panel E inset). Red

*Figure 1 continued on next page*

*Figure 1 continued*

arrowhead (panel E' inset) points to an absence of incorporated monomers at the barbed-end boundary (Z) of the arrays. (F) Schematic representations of the incorporation process. Blue filaments denote previously incorporated ('old') actin, while green (88F) and orange (5C) marks monomers newly incorporated during the indicated pulse. An initial period (0–30 hr APF) of extensive actin polymerization and the establishment of nascent thin-filament arrays are followed by an interim period (30–45 hr APF) of 'patchy' incorporation during which individual, uniform sarcomeric units are defined. The second half of pupal development is devoted to array growth via pointed-end elongation and recruitment of circumferential filaments, as well as turnover at array barbed ends. (G–I) Incorporation patterns of GFP-actin5C (green) following temporally restricted expression pulses using the GAL80^ts/TARGET system. Z-discs are indicated by anti-Zasp52 (red). Restricted expression 'windows' corresponded to 40–90 (G), 50–90 (H) and 70–90 (I) hrs APF. Insets show the GFP-actin5C incorporation patterns in single sarcomeres, in which the prominent Z-disc associated stripe is outlined (Z, Z-disc region; M, M-line region). (J) Quantification reveals a constant width of the GFP-actin5C incorporation stripe overlying the Z-disc region despite the different pulse durations, n = 200 (50 sarcomeres each from 4 different flies). Scale bars correspond to 5 µm in all main panels, 2 µm in the insets.

The following figure supplement is available for figure 1:

**Figure supplement 1.** Peripheral growth of the nascent arrays is continuous.

*Figure 1D,D'* and *Figure 1—figure supplement 1B–F*). This pattern suggests that the initial establishment and structuring of individual sarcomere scaffolds, achieved during this interim phase of pupal development, relies on organization of the existing microfilaments, produced earlier (between 0–30 hr APF). Utilization of newly produced monomers during this interim phase of assembly is limited, and primarily involves 'patchy' incorporation along and (mostly) at the ends of nascent thin-filament arrays, thereby contributing to the generation of uniformly-sized sarcomeres.

A subsequent pulse, between 50 and 90 hr APF, revealed yet a third pattern of actin incorporation (*Figure 1E–E"* and *Figure 1—figure supplement 1G–H"*). During this (final) phase of pupal development, sarcomere units grow noticeably in both length and width, and the characteristic striated pattern of alternating Z-discs and filament-free H zones becomes clearly evident. Newly added actin is observed to form a distinct 'frame' that surrounds a dark rectangular core (insets *Figure 1E and E'*). The core presumably corresponds to the initial thin-filament array assembled during earlier phases. Conversely, the frame-like structure is likely composed of two separate contributions of newly synthesized actin to the nascent core array: extension of the initial fibers at their 'pointed' (M-line associated) ends, and addition of complete new fibers at the circumference of the sarcomere. This picture coincides well with previous studies of global IFM sarcomere growth (*Mardahl-Dumesnil and Fowler, 2001*; *Reedy and Beall, 1993*). Subdividing this relatively large interval further, by initiating GFP-actin88F production at 60 hrs APF, generated similarly shaped 'frames' of monomer incorporation, but of smaller size (*Figure 1—figure supplement 1H–J*). These observations demonstrate that both aspects of actin incorporation- pointed end growth and peripheral thickening- continue throughout the entire period. Monitoring of actin monomer incorporation patterns reveals therefore a dynamic, multi-faceted timeline of IFM thin-filament array assembly during pupal development (*Figure 1F*).

An additional, prominent feature of the GFP-actin pattern following the 50–90 hr APF pulse was a dark stripe overlying the entire Z-disc (insets *Figure 1E'*), implying that actin88F-GFP was not incorporated at this position, in neither the core array nor in newly added filaments at the periphery (*Figure 1E,E"*). This observation suggests that elongation and thickening of the array during the final 50–90 hr APF interval are accompanied by a third mode of monomer incorporation, at the barbed end of the arrays. Absence of GFP-actin88F incorporation at the Z-disc was previously noted by *Roper et al (2005)*, who also demonstrated preferential localization of other GFP-actin isoforms at this site, implying a specific, possibly steric hindrance of GFP-actin88F incorporation. We therefore chose to use the GFP-tagged version of actin5C (*Roper et al., 2005*), a ubiquitous isoform, as a tool for monitoring monomer incorporation dynamics at barbed ends of the thin-filament array. Expression of GFP-actin5C during the 40–90 hr APF time-window resulted in a nearly complementary incorporation profile to the one generated by actin88F-GFP: a prominent stripe of GFP-actin adjacent to the Z-disc and relatively limited incorporation in the peripheral actin strands and pointed ends (*Figure 1G*). Initiation of the GFP-actin5C pulse at different times within this 50 hr interval, generated a thin bright incorporation stripe of constant width adjacent to the Z-disc in all cases (*Figure 1H–J*). This result implies continuous exchange and turnover of actin, rather than actual

growth at the barbed ends of the arrays, and is consistent with the notion that lateral growth of array filaments occurs primarily at their pointed ends (*Littlefield et al., 2001*; *Mardahl-Dumesnil and Fowler, 2001*; *Molnar et al., 2014*).

The correspondence between the regular length of the sarcomeres and the size of the actin monomers can provide a rough estimate of the number of actin monomers that build the entire structure and the proportion undergoing turnover. In a mature sarcomere, the length of thin-filament actin fibers (as measured from the Z-disc to the H-zone) is ~1.70 μm. It is difficult to accurately measure the width of the domain of dynamic actin-monomer exchange due to the limitations of light microscopy resolution, but it is roughly 0.15–0.3 μm on each side of the Z-disc. Given an estimated actin subunit size of 2.7 nm (*Sept et al., 1999*), we can say that a complete thin filament is comprised of ~650 monomers, whereas the zone of continuous exchange at the Z-disc encompasses 50–100 monomers.

## The formin protein Fhos is a major mediator of sarcomeric thin-filament array assembly and growth

To identify actin regulators that are involved in the different phases of IFM thin-filament array organization and growth, we focused on members of the formin protein family, which are major mediators of nucleation and elongation of linear microfilament arrays (*Campellone and Welch, 2010*). The *Drosophila* genome harbors six members of this protein family, each representing a distinct formin subfamily (*Liu et al., 2010*; *Mi-Mi et al., 2012*). To assess their involvement in the IFM sarcomere formation, we used the muscle-specific driver *mef2*-Gal4 to induce expression of RNAi directed against each of the six formins throughout development via UAS-based transgenic constructs, and examined IFM morphology following their isolation from newly eclosed or pharate adults. In most instances, IFM development was only mildly affected, if at all, following individual knockdown of the different *Drosophila* formins.

A severe IFM phenotype was obtained, however, following an expression of RNAi constructs directed against *Fhos*, the single *Drosophila* FHOD sub-family homolog (*Schonichen and Geyer, 2010*). A normally-sized set of six DLM fibers formed in *Fhos* knockdown flies (*Figure 2—figure supplement 1A,B*), indicating that the IFM developmental program is properly initiated. However, the internal organization of these fibers was severely disrupted. This was made apparent by staining the DLMs for key structural components, including α-actinin as a marker for sarcomeric Z-discs, microfilaments and muscle myosin (*Figure 2A–B"*). *Fhos* knockdown DLMs appear to contain myofibril-like elements, but these are thin and randomly oriented (*Figure 2B*). Furthermore, in contrast to the highly regular division of wildtype myofibrils into repetitive sarcomeric units (*Figure 2A,A'*), the abnormally thin *Fhos* knockdown myofibrils display only sporadic α-actinin -stained structures, and a 'smeared', uneven distribution of microfilaments (*Figure 2B,B'*). In addition, muscle-specific myosin is disorganized, and to a large extent lacks an obvious association with microfilaments (*Figure 2B, B"*).

We sought to complement and enhance the analysis of *Fhos*-knockdown IFMs by studying mutant alleles in the *Fhos* locus. A recent study (*Lammel et al., 2014*) described several such alleles, including *Fhos*$^{\Delta1}$, a small deficiency that completely removes the coding regions of most *Fhos* isoforms, and thus represents a severe, possibly null, gene disruption. *Fhos*$^{\Delta1}$ homozygotes die as pharate adults, allowing to assess the effects of *Fhos* gene knockout on IFM development. Immuno-fluorescent staining with informative markers revealed that in *Fhos*$^{\Delta1}$ hemizygous flies that reach the pharate adult stage, DLMs are highly disorganized, lacking even the trace appearance of sarcomeric units observed in *Fhos* knockdown DLMs (*Figure 2C–C"*).

We extended this study by subjecting the mutant DLMs to transmission electron microscopy (TEM) analysis. Longitudinal TEM sections underscored the disorganized nature of the *Fhos* mutant DLMs, which appear to be composed of irregularly shaped myofibrils, lacking a defined spatial orientation (*Figure 2D,E*). While myofilament arrays can be found within these structures, they fail to exhibit any of the features of regularly spaced sarcomeric units characteristic of wildtype DLMs (*Figure 2D*), and display only a few sporadic electron-dense spots that may represent rudimentary Z bands (*Figure 2E*). In contrast to the highly-ordered hexagonal lattice of thick and thin filaments within wildtype myofibrils, revealed by TEM cross-sectional views (*Figure 2F,F'*), the myofilament arrays in *Fhos* mutant DLMs are small, irregularly-spaced, and lack a defined spatial organization (*Figure 2G,G'*).

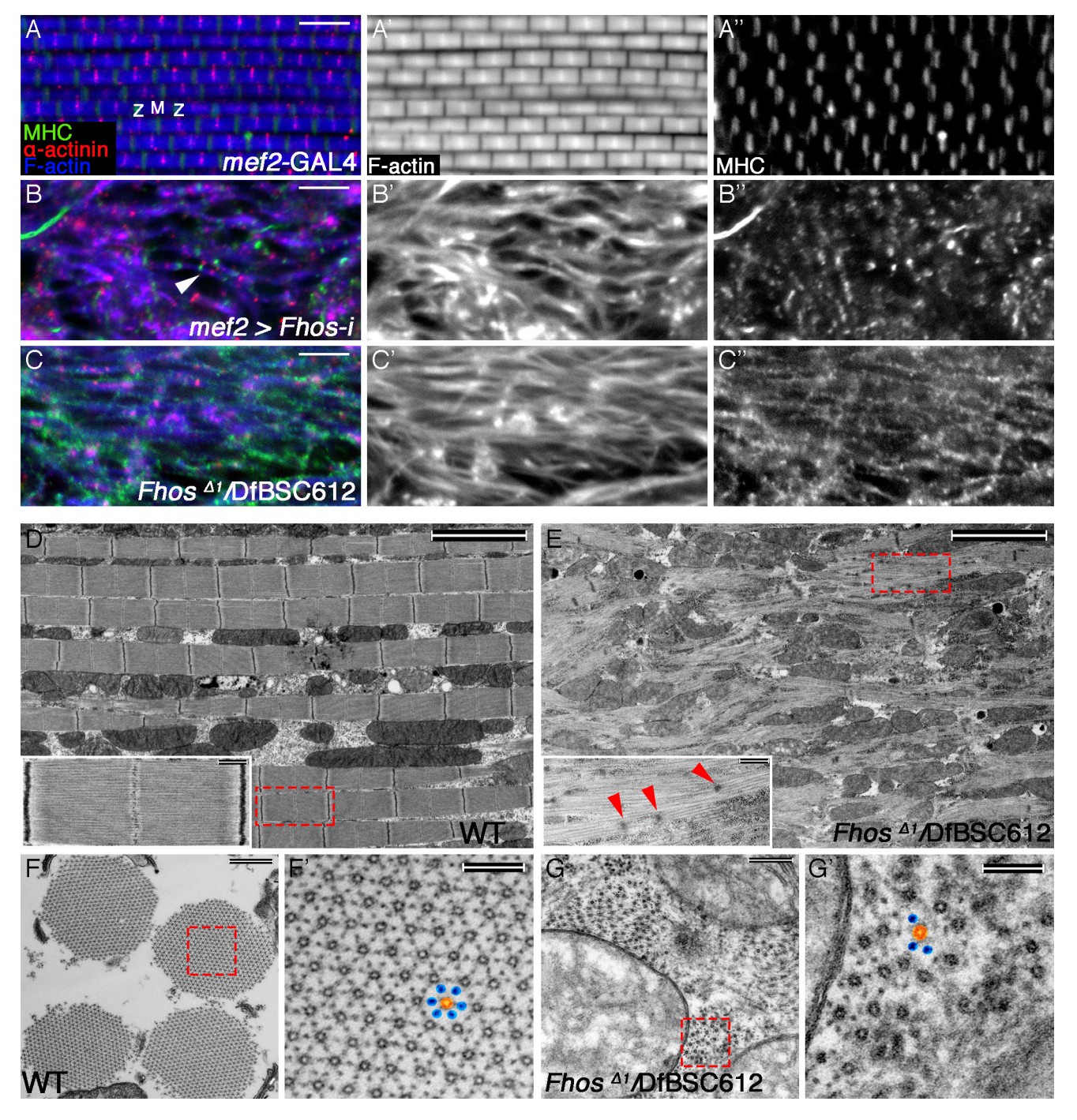

**Figure 2.** The formin Fhos is essential for organization and growth of thin filament arrays. (A–C") Confocal images of IFMs dissected from 1 day old flies or pharate adults and stained with anti- α-actinin (red) to mark Z-disc structures, phalloidin (blue, gray) to visualize microfilaments and anti-MHC (green, gray) to visualize myosin. (A–A") *mef2*-GAL4 control. Z and M mark the Z-disc and M-line of a single sarcomere. (B–B") *mef2*-GAL4>UAS-*fhos* RNAi (knockdown of all *fhos* isoforms). Myofibril and sarcomere structure and organization are defective, but sporadic, undersized sarcomeric units can be observed (white arrowhead in B). (C–C") *fhos*$^{\Delta1}$/Df(3L)BSC612 (*fhos* null). Deletion of the *fhos* locus results in full impairment of myofibril and sarcomeric organization. (D–E) TEM micrographs of longitudinal sections of IFMs dissected from control 1 day old flies (D) and *fhos* null (*fhos*$^{\Delta1}$/Df(3L) BSC612) pharate adults (E). Distinction in the overall myofibril organization is readily apparent, with *fhos* null IFMs lacking typical myofibril and sarcomeric individualization. The insets contrast the stereotypic, highly-ordered structure of the control sarcomeric units (inset D) with the poor organization of arrays within *fhos* null myofibrils and their failure to form individual sarcomeres (inset E). Red arrowheads in (E) point to dispersed, rudimentary Z-discs. (F–G') TEM micrographs of transverse sections of IFMs dissected from control 1 day old flies (F,F') and *fhos* null (*fhos*$^{\Delta1}$/Df(3L)

*Figure 2 continued on next page*

*Figure 2 continued*

BSC612) pharate adults (**G**,**G′**). Primed panels are magnifications of the dashed squares in panels (**F**) and (**G**). In contrast to the highly ordered hexagonal lattice of thick (orange) and thin filaments (blue) in control myofibrils (**F′**), *fhos* null myobrils lack a defined spatial organization (**G′**). Scale bars: 5 μm (**A**-**E**), 500 nm (insets in **D**,**E**, and **F**,**G**), 100 nm (**F′**,**G′**).

The following figure supplements are available for figure 2:

**Figure supplement 1.** Fhos function is required during the early stages of sarcomerogenesis.

**Figure supplement 2.** Fhos is required for proper sizing of thin-filament arrays.

Highly defective myofibril and sarcomere organizations were already clearly apparent in *Fhos*$^{\Delta 1}$ hemizygotes at 50 hrs APF via both light microscopy (*Figure 2—figure supplement 1E–F′*) and TEM *Figure 2—figure supplement 1G–J*) analyses, demonstrating that the mutant phenotypes are a consequence of developmental abnormalities initiating at the onset of IFM sarcomere formation, rather than deterioration of normally formed structures. The severe phenotypes of *Fhos* knockdown and null mutant pupae demonstrate an essential role for Fhos in the assembly and organization of sarcomeric units within IFMs. The nearly complete lack of sarcomeric organization within myofibrils in the absence of Fhos activity implies a critical requirement for Fhos already at early stages of sarcomere assembly.

Such early arrest in sarcomerogenesis may mask potential requirements for Fhos at later stages of the process. To address this issue, we induced RNAi directed at *Fhos* using the IFM-specific *act88F*-Gal4 driver (*Gajewski and Schulz, 2010*), thereby delaying onset of *Fhos* knockdown to a more advanced phase of IFM development. While such flies emerged from the pupal case, they were flightless. A more detailed examination following isolation of IFMs revealed the establishment of an ordered array of intact, regularly-spaced sarcomeric units (*Figure 2—figure supplement 2A–B′*). However, these *Fhos* knockdown sarcomeres exhibited significantly shorter widths and lengths than sarcomeres from age-matched controls (*Figure 2—figure supplement 2C,D*), implying a requirement for Fhos in the elongation and peripheral thickening mechanisms underlying thin-filament array maturation.

Taken together, the range of phenotypic abnormalities associated with the various forms of disruption to Fhos function suggest that Fhos is a major mediator of sarcomere formation, contributing throughout pupariation to different aspects of thin-filament array assembly and maturation.

## Localization of Fhos to Z-discs is critical for function

The *Fhos* locus is composed of two classes of transcripts, utilizing different promoters (*Figure 3A*). Both transcript classes share a set of 3′ exons, but differ in their 5′ regions, which include distinct non-coding and coding exons. As a result, the locus generates two main Fhos protein isoforms, in which a conventional FHOD-family formin, containing all of the canonical formin regulatory and actin-related functional domains, is appended to different N-terminal segments (*Figure 3B*). The isoform encoded by transcripts RA-RG features a short (63 residue) N-terminal domain, while transcripts RH-RJ encode an isoform bearing distinct and substantially larger N-terminal region that is conserved among Neopteran winged-insects (*Bechtold et al., 2014*).

The short Fhos variant, represented by form RA, has been shown to rescue *Fhos* null mutant flies to adult viability, and to restore normal function in affected tissues (e.g. macrophage motility and wing inflation), when expressed ubiquitously via *armadillo*-GAL4 (*Lammel et al., 2014*). We were therefore surprised to discover that the flight muscles of such EGFP-Fhos-PA-rescued flies continued to exhibit severe *Fhos* mutant phenotypes (*Figure 3C–D′*), implying that the short form of Fhos, which mediates most Fhos developmental functions, is insufficient in this context. These observations raised the possibility that the larger isoforms, which have not been extensively studied, provide Fhos activities necessary for IFM sarcomere formation.

To address this issue directly, we utilized a transgenic RNAi construct specifically targeting the large 5′ coding exon (*Figure 3A*). Expression of this RNAi construct in muscle cells, which should eliminate only the long isoforms in this tissue, led to a strong disruption of sarcomere organization, closely resembling null mutations (*Figure 3E,E′*; see also [*Schnorrer et al., 2010*]). Furthermore, we

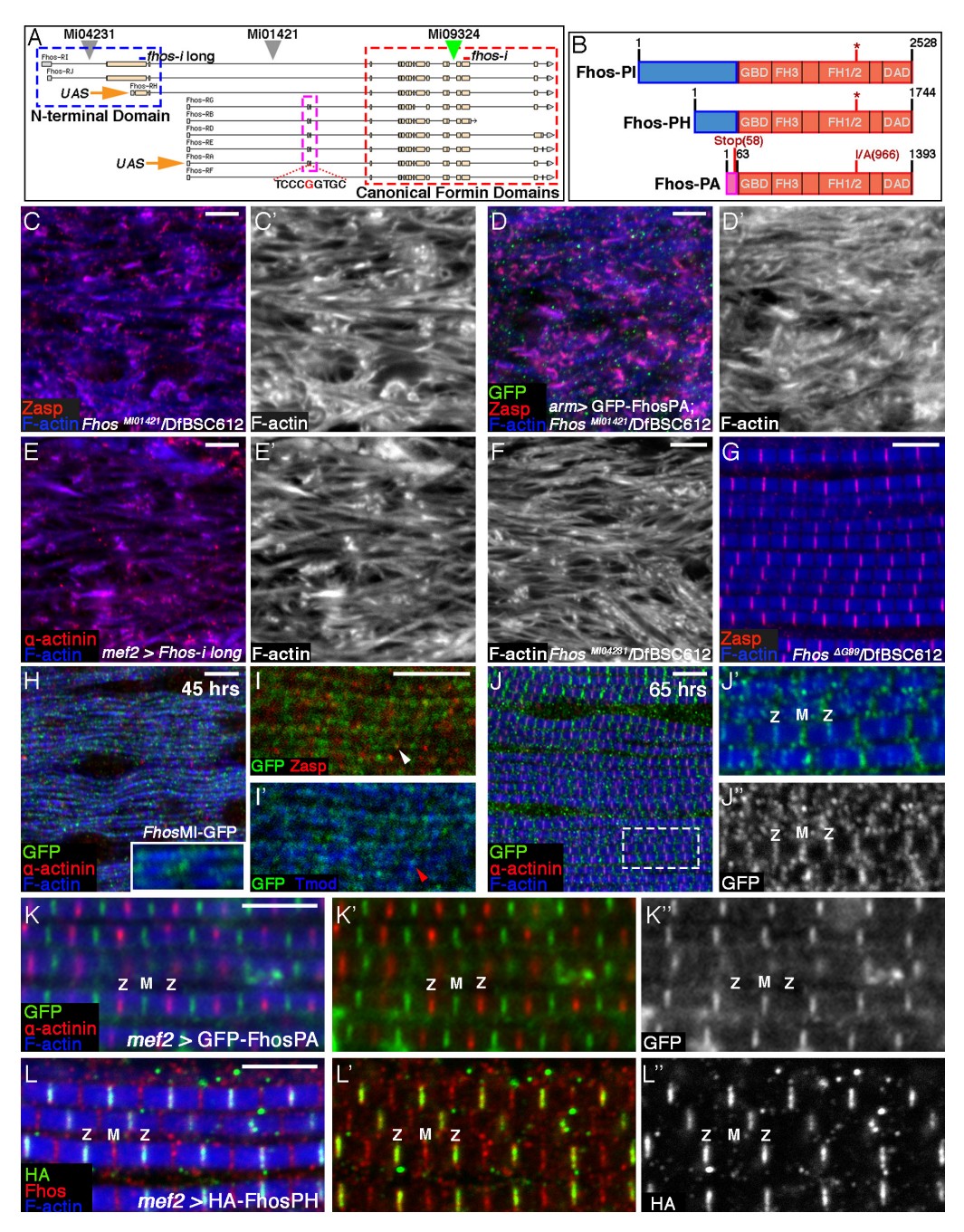

**Figure 3.** Localization of Fhos at the Z-disc is essential for its function. (A) Map of the *Fhos* genomic locus and transcripts (after Flybase, (***Attrill et al., 2016***). Shown are the nine known *fhos* transcripts (designated Fhos-RA–Fhos-RI), which are divided into two groups (RA-RG and RH-RI). The two groups share a nearly identical set of 3' exons (red dashed rectangle), which encode a conventional FHOD-family formin, but are expressed via distinct regulatory regions, and possess distinct sets of 5' exons, including 5' coding exons (blue and purple dashed rectangles) that encode different N-terminal domains. The two transcript variants, RH and RA, used to generate, respectively, the long and short transgenic UAS-Fhos constructs are indicated by orange arrows. The insertion positions of three MiMIC elements, MI04231 (inserted downstream of long isoform initiation sites), MI01421 (inserted downstream of all transcript initiation sites), and MI09324 (used to produce the Fhos-GFP 'protein trap') are indicated by inverted triangles. Positions of two dsRNA target sequences used, one common to all *fhos* isoforms (red bar) and the other specific to the long forms (blue bar) are shown above the transcript map. The CRISPR/Cas9-generated deletion of the guanine residue at position 99 of the short isoform transcript and its adjacent sequence are indicated. (B) Schematic representation of three representative Fhos protein isoforms. The canonical formin domains common to all forms are colored red, while the alternative N-terminal domains are in blue (long forms) and purple (short form). Canonical domains indicated include the GTPase binding domain (GBD), formin homology (FH) domains 1/2 and 3, and the diaphanous autoregulatory domain (DAD). The positions of the

*Figure 3 continued on next page*

*Figure 3 continued*

I966A point mutation in the FH2 domain and the premature stop codon, generate by the frameshift mutation ΔG99 in the Fhos-PA N-terminal domain are indicated. (**C–D'**) Zasp (red) and phalloidin (blue and gray) stainings demonstrate the severe, null-like disruption of myofibril and sarcomere microfilament organization in hemizygous *Fhos^MI01421^*/Df(3L)BSC612 pharate adult flies (**C,C'**), similar to that observed in *fhos^Δ1^* hemizygotes. No rescue is observed following expression of UAS-GFP-Fhos-PA (green) driven by *arm*-Gal4 in this background (**D,D'**). (**E–F**) α-actinin (red) and phalloidin (blue and gray) stainings demonstrate the severe, null-like phenotypes following specific RNAi mediated knockdown of the Fhos long-isoforms (**E,E'**) and in hemizygous *Fhos^MI04231^*/Df(3L)BSC612 (**F**) pharate adult flies. (**G**) Zasp (red) and phalloidin (blue) stainings demonstrate normal myofibril and sarcomeric structure of *Fhos^ΔG99^*/Df(3L)BSC612 hemizygotes, in which the short Fhos isoforms are not expressed. (**H–L''**) Fhos localization in myofibrils, as monitored at two distinct pupal developmental time points, 45 hr APF (**H–I'**), and 65 hr APF (**J–J''**), via a GFP 'exon trap' engineered at the insertion site of the MiMIC transposon MI09324 (green triangle in **A**). The GFP-tagged Fhos proteins (all isoforms) generated in this manner are visualized with anti-GFP (green or gray), Z-discs are visualized with anti-α-actinin or anti-Zasp (red), thin filament pointed ends visualized by anti-Tmod (blue) and microfilaments with phalloidin (blue). The diffuse/punctate initial localization of Fhos-GFP overlying broad portions of the growing myofibrils (**H**), in some cases shows an adjacent localization to the nascent Z-disc (**I** white arrowhead) or to array pointed ends (**I'** red arrowhead). The initial punctate localization gives way to a striated pattern restricted to the vicinities of both the barbed (Z) and pointed (M) ends of the thin-filament arrays (**J–J''**). (**K–K''**) Localization of the short isoform of Fhos in IFMs from a young adult fly, visualized by expression of UAS-GFP-Fhos-PA using the *mef2*-GAL4 driver (anti-GFP, green or gray). Z-discs are visualized with anti-α-actinin (red), and microfilaments with phalloidin (blue). GFP-Fhos-PA localizes to the vicinity of the pointed ends of the arrays (M). (**L–L''**) Localization of the long isoforms of Fhos in IFMs from a young adult fly, visualized with anti-HA (green and gray), following expression of UAS-HA-Fhos-PH using the *mef2*-GAL4 driver. Fhos-PH-PA localizes to the vicinity of the barbed ends of the arrays (Z), where it overlaps with the general Fhos distribution to both the barbed and pointed ends (M) of the arrays (visualized with anti-Fhos [red]). Microfilaments visualized with phalloidin (blue). Scale bars in all panels correspond to 5 μm.

The following figure supplement is available for figure 3:

**Figure supplement 1.** Fhos localization at the Z-disc is essential for its function.

observed similar deleterious effects on the IFM organization in flies hemizygous for MI04231 (*Figure 3F*), a MiMIC insertion allele that is predicted to specifically disrupt the long Fhos isoforms and leave the short isoforms intact (*Figure 3A*). These results lead us to conclude that the long N-terminal protein domain is critical for IFM function of Fhos.

To determine whether the long Fhos isoforms are sufficient, we used a CRISPR/Cas9 approach to generate small deletions in the exon encoding the N-terminal region of the short Fhos isoforms. One of these, *Fhos^ΔG99^*, results in a single nucleotide deletion, generating a translational frameshift and a predicted termination of translation of the short Fhos isoform after only 58 residues (*Figure 3A,B*). *Fhos^ΔG99^* hemizygous flies, which do not express functional short Fhos isoforms, are fully viable and fertile, and do not exhibit any obvious morphological defects. Importantly, the IFMs of these flies display normal myofibril and sarcomere organization (*Figure 3G*). These observations imply that the long Fhos isoforms are sufficient for proper IFM myogenesis, and furthermore, that they can provide most, if not all, functional requirements for *Fhos*.

Identifiable protein domains are not found within the 1198 residue long protein sequence encoded by the large 5' exon, which raised the possibility that this extension to the canonical FHOD-like formin provides a localization cue, rather than an additional functional moiety. To pursue this notion, we set out to determine the localization patterns of the different Fhos isoforms. We first made use of a MiMIC transposable element insertion in the *Fhos* gene locus (MI09324) and the RMCE technique (*Venken et al., 2011*) to generate a GFP 'protein trap' (Fhos-GFP), so that all iso-forms of endogenous Fhos would also harbor a GFP tag (*Figure 3A*). Monitoring the Fhos-GFP signal at 45 hr APF revealed an initial diffuse localization to myofibrils, with some enrichment over nascent sarcomeric units (*Figure 3H–I'*). At later stages (65 hr APF), the localization of Fhos refines to discrete stripes overlying the barbed and pointed ends of the microfilament arrays within the sarcomere (*Figure 3J–J''*), a pattern that persists throughout pupal stages and is still observed in young adult flies (*Figure 3—figure supplement 1A–A'''*). Staining with an antibody we raised to the Fhos C-terminal domains shared by all isoforms of the protein, displayed a similar pattern (*Figure 3—figure supplement 1A–A'''*).

We next examined the localization patterns of representative tagged versions of the short and long isoforms of Fhos, following their expression in IFMs. EGFP-Fhos-PA, representing the shorter isoform, was found to localize exclusively to the vicinity of sarcomere M-lines, corresponding to the pointed-ends of the thin-filament arrays (*Figure 3K–K''*). This observation raised the possibility that

the short isoform is not functional on its own in IFMs due to its restricted localization pattern, which does not include the Z-disc associated barbed-ends of the thin-filament arrays. Notably, instances of ectopic localization of EGFP-Fhos-PA to Z-discs, which were sporadically observed when this construct was over-expressed in a null *Fhos* mutant background, were associated with markedly improved organization of the affected sarcomeres (*Figure 3—figure supplement 1B–B'''*). This observation supports the notion that localization of Fhos to the Z-disc region is critical for its function in the growing sarcomere.

To monitor localization of the long Fhos isoform, we generated an HA-tagged version of Fhos-PH, which contains 410 residues of the novel N-terminal domain (*Figure 3B*). Remarkably, HA-Fhos-PH was found to localize exclusively to the Z-disc region of IFM sarcomeres following expression via *mef2*-GAL4 (*Figure 3L–L''*), in complementary fashion to EGFP-Fhos-PA. Taken together with the genetic analysis, which identified the long isoform as the functional Fhos variant in IFMs, we conclude that localization of Fhos to the Z-disc/array barbed-end region, mediated by the long, novel N-terminal domain, is critical for its sarcomeric function. Interestingly, despite the localization of the HA-Fhos-PH construct to the Z-disc region, it only partially rescued the sarcomere organization defects of *Fhos* null flies (*Figure 3—figure supplement 1C–C'''*). In addition, we observe that the long Fhos isoforms, which provide full sarcomeric functionality, localize to both the Z-disc and M-line regions in *Fhos*$^{\Delta G99}$ hemizygotes (*Figure 3—figure supplement 1D*). Thus, while Z-disc localization of Fhos is an essential requirement for proper sarcomere assembly, we cannot rule out that Fhos performs functional roles at additional sites within maturing IFM sarcomeres.

## The roles of Fhos in actin incorporation into thin-filament arrays

Having established that Fhos is a major contributor to IFM sarcomere organization, we now sought to elucidate its specific roles, by monitoring actin monomer incorporation patterns in the absence of Fhos function. Towards this end, we examined IFMs from *Fhos* knockdown pupae, following a restricted expression of GFP-actin88F during early (0–30 hr APF), interim (30–60 hr APF) and late (60–90 hr APF) phases of pupal development (*Figure 4A*). During the initial stages of IFM development, incorporation of GFP-actin88F into unstructured microfilament arrays within nascent myofibrils proceeded normally in *Fhos* knockdown IFMs (*Figure 4B,C*). This finding implies that Fhos is not essential for the initial 'burst' of strong polymerization activity characteristic of this phase (*Figure 1C–C''*), and is consistent with the establishment of properly sized but internally disorganized DLM myofibers in *Fhos* knockdown and mutant flies (*Figure 2*, *Figure 2—figure supplement 1*).

In contrast to these observations, a marked effect of *Fhos* knockdown on the actin monomer incorporation pattern can be discerned during the interim (30–60 hr APF) period of pupal development (*Figure 4D–G*). While wildtype IFMs display an irregular 'patchy' pattern of incorporation spread out over the nascent sarcomeric arrays (*Figure 4D–E*), *Fhos* knockdown IFMs exhibited a repetitive, undulating pattern, with peaks of incorporation centered at the pointed-ends of the arrays (*Figure 4F–G*). Fhos therefore appears to mediate monomer incorporation into filament patches, but is not involved in the emerging process of array elongation from pointed ends. This feature of *Fhos* knockdown IFMs persists during the final phase of pupal development, when wildtype sarcomeres display a frame-like pattern of incorporation (*Figure 4H–J*). While such 50–90 hr old *Fhos* knockdown IFMs retain pointed-end incorporation, they appear to be thinner and lack the circumferential accumulation of incorporated GFP-actin88F, which represents radial growth of the arrays through addition of peripheral microfilaments (*Figure 4I,J*), implying a requirement for Fhos in the array 'thickening' process. Finally, a GFP-actin5C incorporation band of normal width was readily detected adjacent to sarcomere Z-discs in *Fhos* knockdown IFMs (*Figure 4K–M*), implying that actin monomer exchange at barbed ends of the arrays is Fhos independent.

Analysis of actin monomer incorporation patterns thus provides a higher resolution and reveals specific roles for Fhos in mediating thin-filament array assembly and growth. These include the organization of microfilaments into nascent structures, shaping sarcomeres into uniformly sized and regularly-spaced units and, finally, radial growth of the arrays via peripheral thickening. On the other hand, the synthesis of the initial pool of microfilaments, array elongation from pointed ends and actin exchange at the Z-disc do not require Fhos, and thus rely on the activity of other actin regulators.

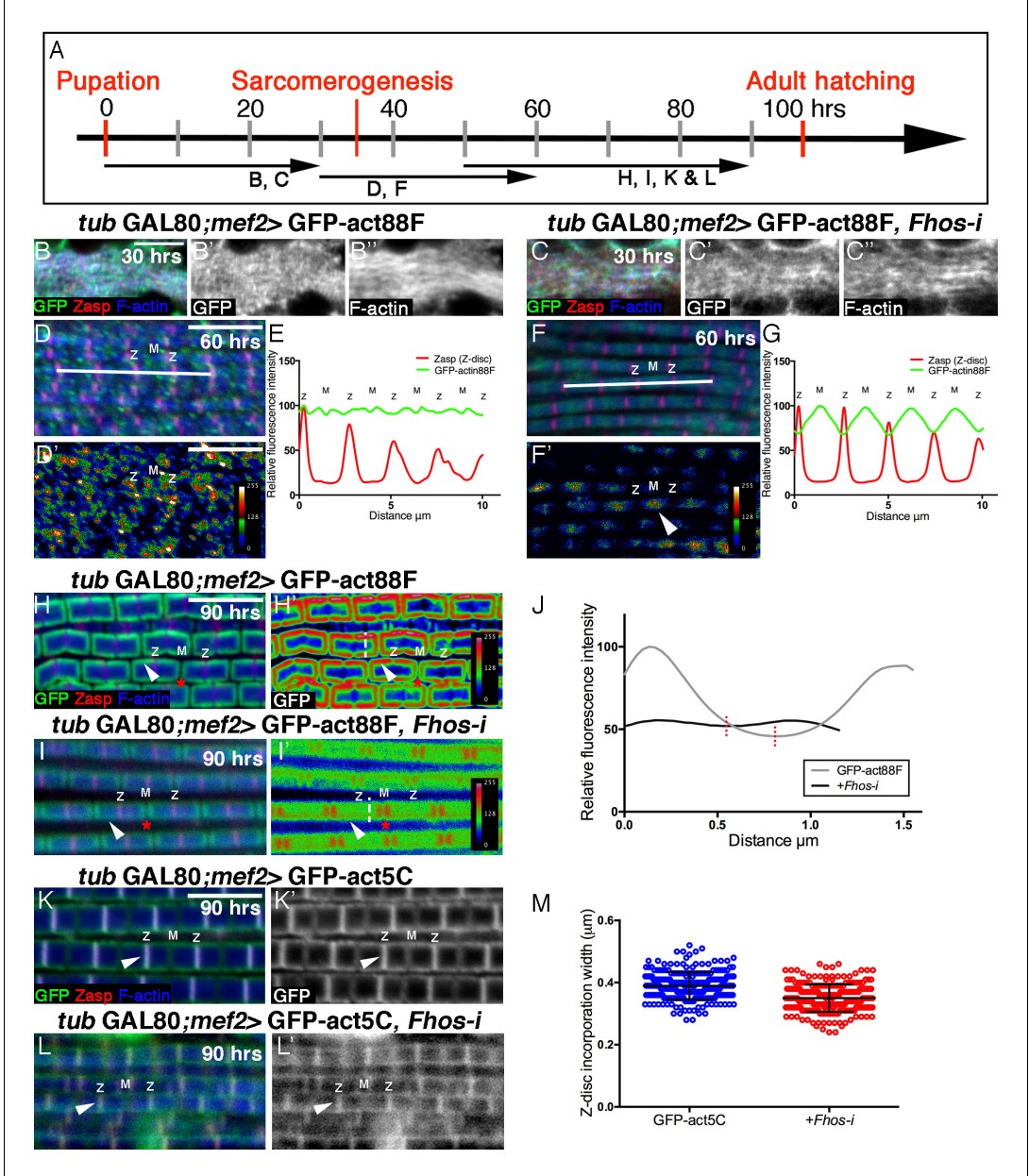

**Figure 4.** Fhos is required for the 'patchy' actin monomer incorporation and radial expansion aspects of thin-filament array growth. (**A**) Scheme of IFM development intervals used for temporally restricted expression of GFP-actin88F (**B–J**) or GFP-actin5C (**K–M'**) in wildtype (**B–B″,D,D',H,H',K,K'**) or *Fhos* knockdown (**C–C″,F,F',I,I',L,L'**) IFMs. (**B-I'**) GFP-actin88F (green, gray) expression between 0–30 (**B–C'**), 30–60 (**D-G**) and 50–90 (**H-I'**) hr APF. Z-discs are visualized with anti-Zasp52 (red) and microfilaments with phalloidin (blue, gray). (**B–C'**) The general and uniform incorporation of monomers into microfilaments characteristic of the initial phase of sarcomere formation (**B–B″**) is not affected in *fhos* knockdown myofibrils (**C–C″**). (**D–G**) The dispersed and 'spotty' wildtype incorporation pattern during the interim (30–60 hr APF) phase (**D**) is replaced by a pointed-end centered pattern in *fhos* knockdown myofibrils (**F**). These pattern distinctions are further demonstrated by heat maps of the GFP-actin88F distribution (**D',F'**) and quantification of GFP intensity (**E,G**) derived from 10 μm profiles covering approximately four sarcomeric units (white lines in D and F; data were acquired for 15 profiles from 7 different pupae for each genotype [n = 105]). (**H–I'**) The incorporation 'frames' normally generated by late GFP-actin88F expression pulses (**H,H'**) lack peripheral incorporation (arrowheads) following *fhos* knockdown (**I,I'**), but these abnormally thin myofibrils retain proper incorporation at array 'pointed' ends (red asterisks). The lack of incorporation 'frames' is also demonstrated by heat maps (**H',I'**), and the quantification of GFP intensity (**J**; data were acquired for 100 profiles from 4 different flies for each genotype [n = 400]) along vertical profiles (white dashed line **H',I'**), which show the loss of peripheral incorporation and thinner sarcomeres (red dashed lines in **J**). (**K-M**) GFP-actin5C (green, gray) expression between 50–90 hr APF. Z-discs are visualized with anti-Zasp52 (red) and microfilaments with phalloidin (blue). GFP-actin5C induction in parallel to *fhos* knockdown showed normal Z-disc associated turnover (arrowheads). (**M**) Quantification of Z-disc incorporation band width (data were acquired for 50 Z-discs from 4 different flies for each genotype [n = 200]). Scale bars in all panels correspond to 5 μm.

## Sals is required for thin filament elongation from the pointed ends of the array

As the incorporation data suggests that filament extension from the pointed-ends of the arrays does not require Fhos, we sought to identify alternative elements that may mediate this key aspect of sarcomere growth. Such factors are unlikely to include formin family members, since formins are primarily thought to extend actin filaments from their barbed ends (*Campellone and Welch, 2010*; *Goode and Eck, 2007*). The WH2-domain protein Sarcomere length short (Sals) is an attractive candidate, since it was shown to contribute to pointed-end filament elongation in *Drosophila* larval muscles, and to be localized to the pointed ends of thin-filament arrays in IFM sarcomeres (*Bai et al., 2007*).

We first assessed the role of Sals in IFM sarcomerogenesis by using the *mef2*-GAL4 driver and the GAL80[ts]/TARGET system to express an RNAi construct targeted against *sals* from the onset of pupariation, and examined the effect on IFMs of pharate adult flies. The overall length of the *sals*-knockdown thin-filament arrays is significantly shorter than control arrays, and their pointed end borders are abnormally shaped and discontinuous (*Figure 5A–C'*, *Figure 5—figure supplement 1D*), consistent with a pointed-end extension function for Sals.

We next utilized the GFP-actin88F incorporation assay to further examine Sals activity in developing IFMs. Specifically, *sals* function was disrupted by knockdown beginning at 50 hr APF, when 'core' thin-filament arrays have already formed, monomer incorporation was monitored in young adults. The resulting incorporation pattern is a near 'mirror-image' of *Fhos* knockdown during this period: shortened sarcomeres displaying normal peripheral thickening of the core arrays (*Figure 5D–E',G*), coupled with a significant decrease of actin incorporation at their pointed ends (*Figure 5F*).

The conserved pointed-end capping protein Tropomodulin (Tmod) (*Gregorio and Fowler, 1996*) is a second factor that could contribute to filament pointed-end elongation. Induction of RNAi targeting *tmod* at 0 hr APF indeed results in a significant shortening of IFM thin-filament arrays, but to a considerably lesser extent than the shortening observed following knockdown of *sals* (*Figure 5—figure supplement 1A–D*). Furthermore, *tmod* knockdown during the second half of pupal development only weakly affects the array length and pointed-end incorporation of actin monomers (*Figure 5—figure supplement 1E–H*).

Taken together, these results identify Sals as a major actin regulator, specifically mediating pointed-end growth of thin-filament arrays throughout IFM maturation, while the contribution of Tmod appears to be restricted to early stages of IFM development.

## The barbed-end associated capacities of Fhos are dispensable during the initial stages of sarcomere organization

Formins employ a variety of molecular mechanisms for regulating microfilament organization and dynamics, including microfilament nucleation, elongation, bundling and capping activities (*Goode and Eck, 2007*; *Harris and Higgs, 2006*; *Schonichen and Geyer, 2010*). To begin to address this issue in the context of Fhos IFM function, we used CRISPR/Cas9 technology to insert a point mutation into the endogenous *Fhos* locus, thereby generating a single amino acid substitution (I966A according to residue numbering of the short form of Fhos, *Figure 3B*). Mutating this highly conserved residue in a variety of formins, including the mammalian Fhos homolog FHOD3, consistently abolished activities requiring barbed-end association (actin nucleation, elongation and capping) (*Harris et al., 2006*; *Taniguchi et al., 2009*; *Xu et al., 2004*).

Fhos[I966A] hemizygous flies are viable and exhibit an externally normal morphology, implying that the nucleation activity of Fhos is generally dispensable. However, the IFMs of these flies contain abnormally thin myofibrils. Most of these myofibrils display, nevertheless, an organized structure of repeated sarcomeric units, with clear demarcation of the Z-discs (*Figure 6A–A"*), suggesting an arrest in sarcomere growth following proper initial assembly and organization. This notion was further borne out following TEM-level visualization, which showed that the thin Fhos[I966A] myofibrils house properly structured and evenly-spaced sarcomeres (*Figure 6B*), harboring well-ordered lattices of thick and thin filaments (*Figure 6C*).

The actual radial size of Fhos[I966A] sarcomeres was assessed by determining the number of thick filament units in myofibril TEM cross sections. This analysis revealed that the sarcomeres of 1 day old

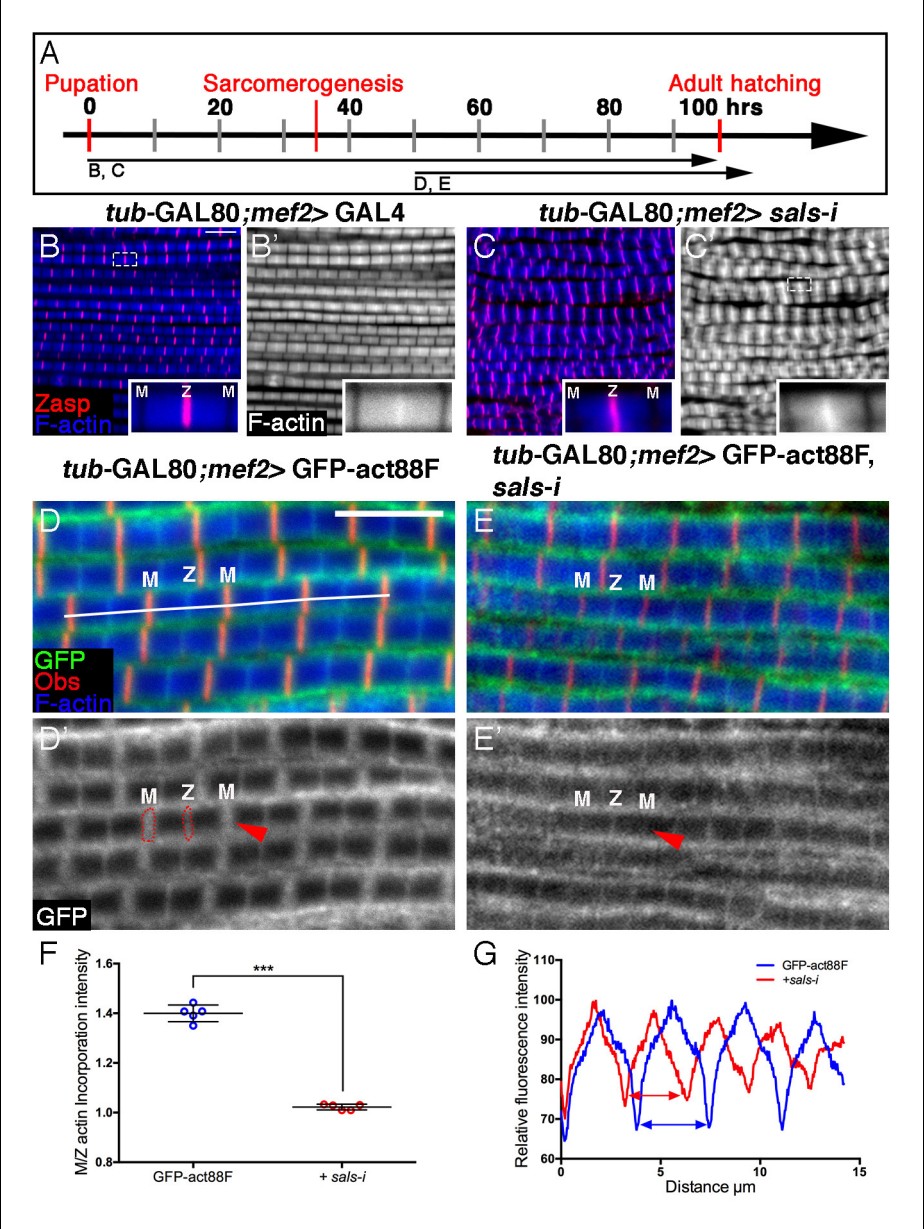

**Figure 5.** Sals is required for 'pointed-end' thin-filament growth. (**A**) Scheme of IFM development intervals used for temporally restricted expression of *sals* RNAi and GFP-actin88F. (**B–C'**) IFMs dissected from control (*mef2*-GAL4) 1 day old flies (**B,B'**) and *sals* knockdown pharate adults (**C,C'**), in which RNAi expression was initiated at 0 hr APF. Z-discs are visualized with anti-Zasp52 (red) and microfilaments with phalloidin (blue or gray). s*als* knockdown results in sarcomere shortening and 'pointed' end abnormalities (insets in **B'** and **C'**; for quantification see *Figure 5—figure supplement 1D*). (**D–E'**) IFMs dissected from young (1–2 day old) flies in which GFP-actin88F expression (anti-GFP, green, gray) was initiated at 50 hr APF on its own (**D,D'**) or together with *sals* RNAi (**E,E'**). M lines are visualized with anti-Obscurin (red) and microfilaments with phalloidin (blue). (**F**) The GFP-actin88F incorporation band at the 'pointed' ends is significantly decreased following *sals* knockdown, as shown by the M/Z intensity ratio (p<0.0001, P values determined by Mann-Whitney test), while the addition of peripheral microfilaments is unaffected. (**G**) Quantification of phalloidin intensities derived from 13 μm profiles (white line in **D**). *sals* RNAi myofibrils (red line) exhibit shorter sarcomeric units compared to control (blue line). Scale bars in all panels correspond to 5 μm.

The following figure supplement is available for figure 5:

**Figure supplement 1.** Involvement of Tmod in nascent thin filament array elongation.

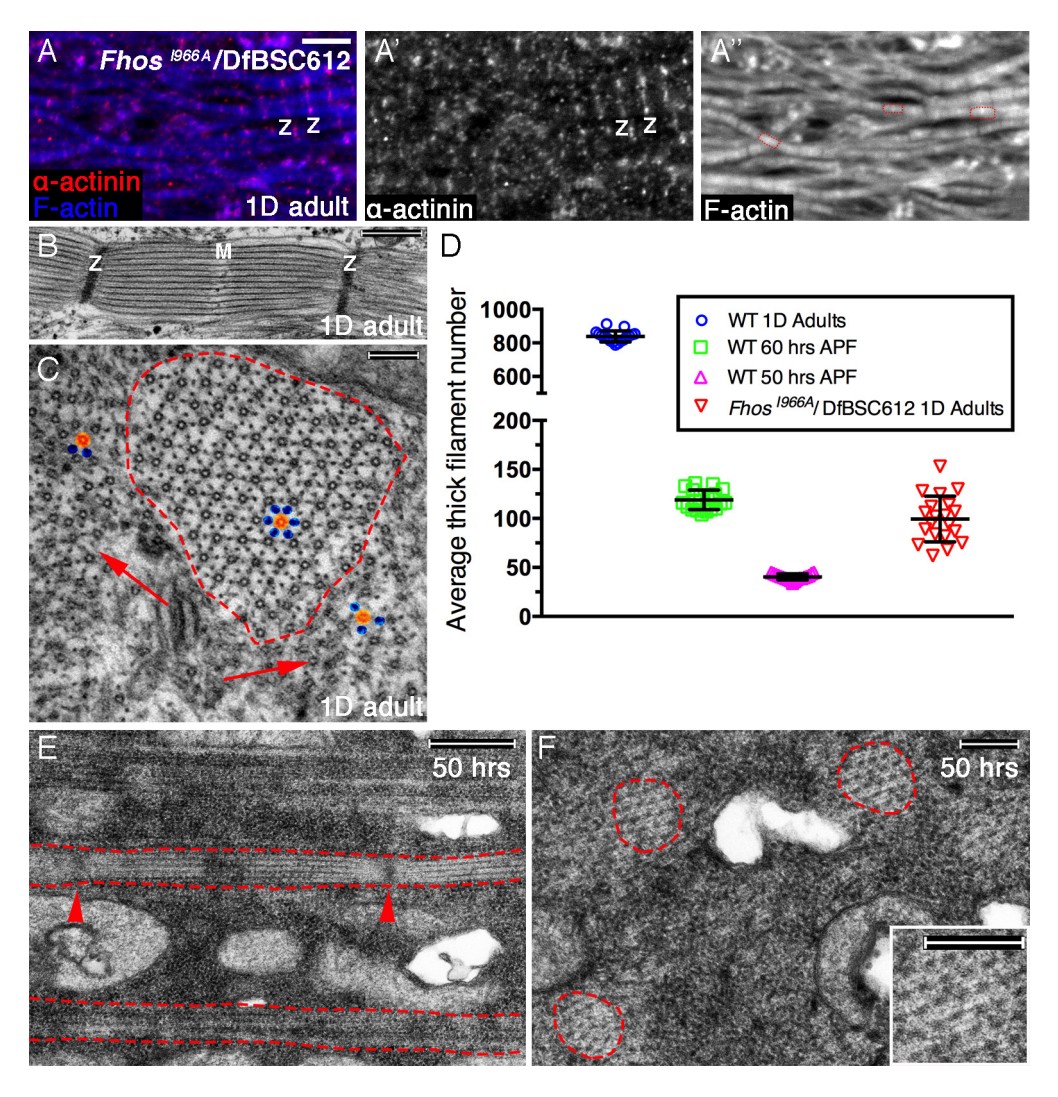

**Figure 6.** Early Fhos function does not require barbed-end activities. (**A–A''**) IFM myofibrils from a *Fhos*[I966A]/Df(3L) BSC612 1 day old adult fly. Z-discs are marked by anti-α-actinin (red or gray) and microfilaments are visualized with phalloidin (blue or gray). The thin myofibrils display organized arrays of repeated sarcomeric units (red outlines in **A''**). (**B–F**) TEM analysis of IFM myofibrils from *Fhos*[I966A]/Df(3L)BSC612 flies. (**B,C**) One day old adult flies. A longitudinal section (**B**) shows a stereotypic sarcomeric unit displaying clear Z-disc (Z) and M line (M) structures. A cross section (**C**) shows an individual sarcomeric unit (red dashed circle) harboring a well-formed lattice of thick and thin filaments. Arrows point to accumulations of nearby filaments, which could serve as a source for radial growth, but have not been recruited. The degree of lattice organization within and outside the sarcomere can be appreciated from the spatial arrangement of representative thick (orange) and thin (blue) filaments. (**D**) Quantification of sarcomere size in wildtype and *Fhos*[I966A]/Df(3L)BSC612 pupae and adult flies at the indicated ages, based on the number of thick filament units in TEM cross-sections (n = 20 for each background). (**E,F**) 50 hr APF pupae. Longitudinal (**E**) and cross (**F**) sections show myofibril individualization (red dashed lines in **E**) and formation of nascent sarcomeric units (red dashed circles in **F**) with defined Z-disc borders (red arrowheads in E; see also **Figure 2—figure supplement 1G,H**). Scale bars correspond to 5 μm (**A**), 100 nm (**B**), 500 nm (**C,F**) and 200 nm (**E**).

Fhos-I966A mutants are ~ 8 times smaller than wildtype sarcomeres from similarly aged flies, and correspond in size to normal sarcomeres between 50–60 hr APF (*Figure 6D*). This observation is in keeping with the notion of an arrest in sarcomere growth during intermediate pupal stages. Furthermore, the appearance and organization of sarcomeres within myofibrils of Fhos[I966A] hemizygotes at

50 hr APF (*Figure 6E,F*) closely matches those of sarcomeres from age-matched wildtype pupae (*Figure 2—figure supplement 1G,H*).

These light and electron microscopy analyses suggest therefore that the normal involvement of Fhos at the initial stages of IFM sarcomere assembly does not require a barbed end-associated activity, consistent with the early diffuse localization pattern of Fhos in myofibrils. However, the barbed-end localization and associated activities of Fhos appear to be essential for the maturation of nascent arrays, implying a multi-faceted involvement of this formin in sarcomere formation.

## Discussion

Formation of the adult *Drosophila* IFMs during pupariation provides an established model system to study formation of skeletal muscles, and in particular the generation of the repeated sarcomere structure, the core functional unit underlying muscle contractility. The IFM system possesses several key features that make it amenable for detailed analysis. These include an extended developmental time window of ~60 hr, the availability of genetic methods for investigation (e.g., RNAi-based knockdown) and the highly ordered and repetitive organization of sarcomeric units within IFM myofibrils, which allows for the detection and analysis of both major and subtle alterations and defects. Our study focused on the actin based thin-filament array component of sarcomeres. Towards this end, we utilized an additional, highly useful feature of the IFM system: the capacity to monitor the incorporation pattern of new actin monomers at discrete stages of the process, using transgenic GFP-actin constructs whose expression can be readily manipulated. The analyses performed using the various approaches and tools available for the study of the IFMs allow us to chart the principles, timeline and molecular basis for assembly, organization and maturation of thin-filament arrays within this model of sarcomerogenesis.

While it is likely that regulators of linear actin belonging to the formin family play a central role in the formation and maintenance of actin filament arrays in the sarcomere, redundancy among them appears to be commonplace in this context (*Mi-Mi et al., 2012*; *Rosado et al., 2014*), complicating the elucidation of distinct functional roles. Our detection of severe sarcomeric phenotypes following disruption of *Fhos* activity on its own, identifies Fhos as a key contributor to the processes governing IFM thin-filament array assembly, and sets the stage for studying the involvement of formins in this major aspect of microfilament organization, via utilization of genetic approaches. FHOD-family formins have been previously associated with sarcomere organization in cardiac muscle (*Iskratsch et al., 2010*; *Kan et al., 2012*; *Taniguchi et al., 2009*; *Wooten et al., 2013*), but the roles these proteins play during skeletal myogenesis have been difficult to ascertain. The ability to dissect the program of IFM sarcomere formation and to disrupt the activity of Fhos to different extents and at distinct phases, provides a comprehensive view of the role of this formin-family protein in the process, and of skeletal muscle thin-filament array assembly at large (*Figure 7*).

1. The onset of pupal development is accompanied by a prominent burst of actin polymerization, generating a store of microfilaments for use during subsequent stages, when polymerization activity declines considerably. The identity of actin nucleators and in particular formins involved in the extensive initial polymerization is not known, and it is certainly possible that several formins act redundantly, given our failure to disrupt this process by single formin knockdowns, including that of *Fhos*. An initial sign of internal myofiber organization is seen in the segregation of the abundant microfilaments produced during the early phase of pupal development, into elongated myofibrils that align in parallel to the fiber. We do not know the nature of the signal governing this alignment, but it is noteworthy that previous studies of the dynamic organization of sarcomeres in cultured muscle cells have indicated that microtubules which are aligned along the fiber may provide an initial cue for the recruitment and orientation of myosin heavy chain and possibly microfilaments as well (*Pizon et al., 2005*).

2. The initial, widespread polymerization gives way to a more limited mode of incorporation, characterized by 'patches' of actin monomers that are added to a nascent microfilament array, suggesting that this period is devoted to proper structuring of the arrays within individual, uniformly-sized and regularly separated sarcomeric units. It is during this interim period that disruption of Fhos activity first leads to alteration of the monomer incorporation pattern, in what we interpret to be a telling fashion, as it retains an underlying, highly regular pattern of monomer incorporation at the pointed-ends of the thin filament arrays (*Figure 4D–G*). This observation constitutes a direct

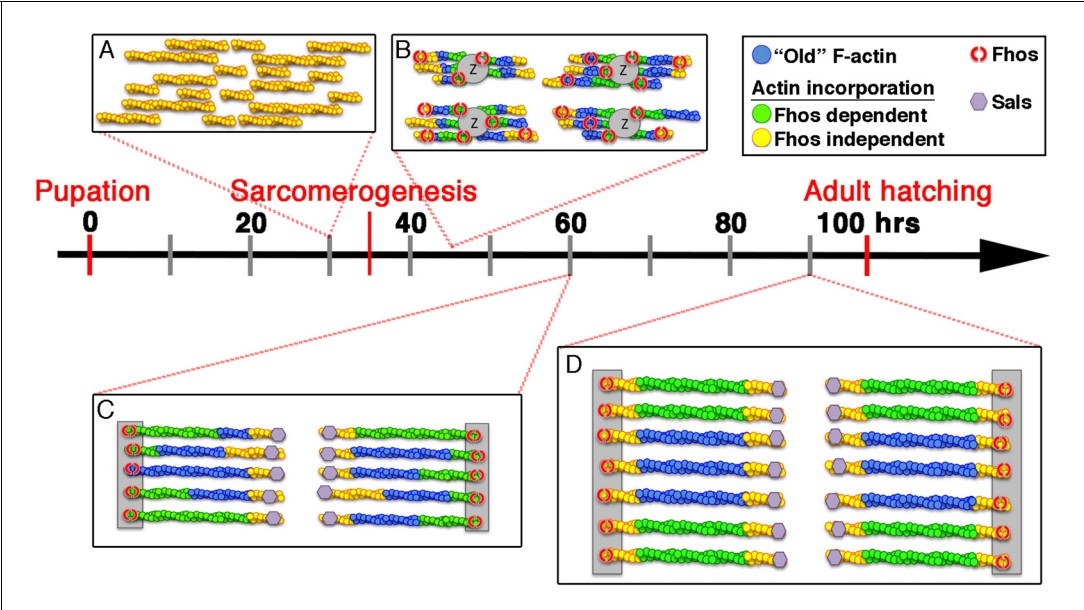

**Figure 7.** Model of IFM thin-filament array assembly and the roles of Fhos and Sals. Four distinct stages of thin-filament array assembly and maturation are represented in correlation to pupal developmental stages. (**A**) Extensive filament polymerization (0–30 hr APF), which takes place in a Fhos-independent manner. (**B**) Organization of nascent sarcomeres and patched actin incorporation (30–45 hr APF). Fhos localizes to the nascent arrays, and is required for their organization into discrete structural units. (**C,D**) Growth and maturation of nascent sarcomeres (45 hr APF- eclosion). Fhos localizes to the vicinity of the Z-discs, and is essential for radial growth of the thin-filament arrays, possibly via peripheral filament recruitment. Two additional actin incorporation modes are executed in a Fhos independent manner. Z-disc associated monomer turnover, and 'pointed' end elongation, which is mediated primarily by Sals.

demonstration that IFM arrays elongate from their pointed-ends, contrary to the conventional barbed end-biased growth of microfilaments, and in keeping with previous findings based on studies of the pointed-end capping protein Tropomodulin in both IFMs and vertebrate cardiomyocytes (*Littlefield et al., 2001*; *Mardahl-Dumesnil and Fowler, 2001*). Furthermore, our investigation identifies the WH2-domain protein Sals as a major mediator of the pointed-end growth of IFM thin-filament arrays, similar to its function in *Drosophila* larval muscle sarcomeres (*Bai et al., 2007*). Interestingly, Leiomodin acts to mediate pointed-end elongation in vertebrate cardiomyocytes (*Chereau et al., 2008*; *Tsukada et al., 2010*), implying a conserved function for WH2-domain actin regulators in sarcomerogenesis.

Interfering with Fhos activity results in severe impairment of myofibril and sarcomere organization (*Figure 2*), raising the question of how to reconcile the strong mutant phenotypes with the limited degree of Fhos-dependent actin incorporation and array growth during the early and interim periods of pupal development. We suggest that, rather than participating directly in the process of actin polymerization, Fhos plays a key organizational role during this period, which leads to the initial structuring of thin-filament arrays, using microfilaments generated by other formins and actin regulators. FHOD-family formins are considered to be poor microfilament nucleators (*Schonichen et al., 2013*; *Taniguchi et al., 2009*), and are thought to act via alternative modes- such as microfilament bundling (*Kutscheidt et al., 2014*; *Schonichen et al., 2013*)- to provide shape and structure to microfilament arrays. Our analysis is consistent with this notion, as we have demonstrated that Fhos-I966A, a Fhos variant presumably lacking barbed end associated activities, supports formation of small but properly organized sarcomeres, similar to the sarcomeres normally formed during the early and interim stages of pupal development. We suggest therefore, that Fhos plays a critical early role, coupling organization of pre-existing filaments into ordered arrays, with a secondary but important capacity to coordinate array size and structure through 'patchy' monomer incorporation.

3. Once the rudimentary sarcomere 'core' is assembled and defined Z-discs with fixed spacing are established, further extension and addition of actin filaments is dictated by the initial

organization of the sarcomere. It is at this stage that Fhos assumes a striated localization pattern corresponding to thin-filament array ends, and where localization to the barbed end region becomes critical for Fhos function. We identified three different modes of actin incorporation which contribute to the nascent sarcomere maturation during the later stages of pupal development:

1. Elongation. Thin-filament arrays continue to extend laterally. Growth occurs from the pointed-ends and is mediated primarily by Sals. On the other hand, no evidence for extension at the barbed ends is evident from actin-GFP incorporation. We suggest that the reduced lateral size of arrays observed following late-stage *Fhos* knockdown may arise from the compromised protection of the barbed ends immediately adjacent to the Z-disc.

2. Radial 'thickening'. The sarcomeres grow radially due to the circumferential addition of actin filaments at their periphery. The added filaments are predominantly synthesized at this later stage, since they are readily labeled by monomeric actin produced only at that time (*Figure 1E,E'*). Radial growth requires Fhos, and therefore constitutes a second major contribution of this formin to thin-filament array assembly. Fhos may contribute to radial thickening by bundling and recruiting complete filaments to the periphery of the array core. The reliance on localization to the vicinity of the Z-discs and the arrest in growth of FhosI966A sarcomeres prior to radial thickening strongly imply that this activity depends on association of Fhos with the barbed ends of the thin-filament arrays.

3. Barbed-end turnover. An unexpected finding was the identification of rapid actin turnover that is highly restricted to the barbed ends of the thin-filament arrays. The barbed-end turnover persists throughout the late phase of pupariation, and does not contribute to filament elongation (*Figure 1G–J*). Interestingly, barbed-end turnover could be detected with the GFP-tagged form of the ubiquitous actin isoform actin5C, but not with the IFM-specific isoform GFP-actin88F. This may reflect a structural constraint that underlies the use of distinct actin isoforms for different aspects of array growth and maintenance.

The contribution of barbed-end turnover to thin-filament array organization is not readily apparent. Turnover may be part of a mechanism that maintains the filament integrity by a continuous process of elongation and disassembly within the dynamic environment of the maturing Z-disc. Turnover does not require Fhos, and its molecular basis is currently unknown. It will be interesting to examine if a similar process is operating in adult muscles, to maintain their integrity in the face of extensive contraction activity during flight (*Perkins and Tanentzapf, 2014*).

In conclusion, our analysis of IFM sarcomeres implies that the assembly of the highly structured thin-filament array is an elaborate, stepwise process involving diverse aspects and machineries of microfilament nucleation, growth and organization. The formin protein Fhos plays a central role, contributing to array assembly at several stages of the process. Fhos acts initially to mediate the assembly of thin-filament arrays within discrete sarcomeric units. Fhos localization to the Z-disc region, an apparently conserved feature among FHOD-family formins (*Iskratsch et al., 2010*; *Mi-Mi et al., 2012*; *Rosado et al., 2014*), becomes essential for function during the later stages of IFM development, where Fhos plays an essential role in sarcomere radial growth. Interestingly, it has been suggested that localization of FHOD3 to the Z-lines of murine cardiomyocytes is a regulated process, relying on phosphorylation of a short domain encoded by an alternatively-spliced exon (*Iskratsch and Ehler, 2011*; *Iskratsch et al., 2010*), echoing the importance of Fhos Z-disc localization described here.

While this study has elucidated specific roles for a FHOD-family formin in a model system of skeletal muscle sarcomerogenesis, key issues remain open. The precise molecular nature of the microfilament-associated activities of Fhos, the mechanistic significance of its spatial localization patterns, regulation of its sarcomeric activities and their coordination with other functional elements of the actin-based cytoskeleton, all await further investigation.

## Materials and methods

### *Drosophila* genetics

GAL4 drivers included *mef2*-GAL4 (*Ranganayakulu et al., 1996*) and *act88F*-GAL4 *Gajewski and Schulz, 2010*], BDSC 38461). UAS-*Dicer2* elements were included for enhancement of RNAi activity (*Dietzl et al., 2007*). GFP-actin lines (*Roper et al., 2005*) included UAS-GFP-*act5C* (BDSC # 9258) and UAS-GFP-*act88F* (BDSC # 9253 and # 9254).

UAS-*dsRNA* lines used: *fhos* (VDRC GD2374 [knockdown of long forms] and VDRC KK108388 [general Fhos knockdown]); *sals* (VDRC KK112869 and TRiP JF01110); *tmod* (VDRC GD32602 and TRiP JF01094)

Crosses were commonly kept at 25°C. Temporally-controlled expression protocols utilized the GAL80$^{ts}$/TARGET system (*McGuire et al., 2004*). F1 progeny were raised at 18°C and shifted to 29°C at 0 hr APF (white pupae). Pupae were then grown at 29°C until the desired developmental time, taking into account the accelerated pupal development (1 hr at 29°C equals approximately 1 hr and 20 min at 25°C) and the time for a complete substitution to inactive form of GAL80 (approximately 5 hr). All indicated time windows are equivalent to the developmental periods for flies grown at 25°C.

Generation of the different fly lines described in the study was achieved as follows:

Fhos-GFP protein trap line: obtained by injection of the protein trap plasmid *Splice phase 0 EGFP-FIAsH-StrepII-TEV-3xFlag*, into the MiMIC insertion line *Fhos*$^{MI09324}$, as described (*Venken et al., 2011*).

UAS-PH-Fhos transgenic line: A full length Fhos-RH cDNA was assembled from clones IP17223 and SD08909 using restriction free cloning as described (*Unger et al., 2010*). The resulting construct was amplified by PCR, cloned into the NotI–KpnI sites of the pUAST–attB vector containing N-terminal 3XHA and injected following sequence verification into an attP40 line to produce transgenic flies.

Short isoform mutant alleles: A CRISPR/Cas9-based approach (*Gratz et al., 2013*) was used to target the 1$^{st}$ coding exon of the short Fhos isoforms. A guide RNA template complementary to sequences within the exon (5'CTTCGCCGCCTTCCCGATCCCGGTG3') was synthesized and cloned into the pU6-BbsI-chiRNA plasmid (Addgene), and the plasmid was injected into *vasa*-Cas9 embryos (BestGene). Lines were established from the progeny of the injected flies. Mutations were identified by sequencing PCR-amplified genomic DNA encompassing the relevant exon from flies bearing candidate mutant 3$^{rd}$ chromosomes. Two deletion events were identified in this manner, a single guanine nucleotide deletion giving rise to *Fhos*$^{\Delta G99}$ (*Figure 3A*) and a four nucleotide deletion (positions 101–104). Both deletions result in translational frameshifts and premature translational arrest at amino-acid position 58 of the protein sequence, and display identical phenotype and localization features.

Primers used included 5' GCGTGGCGTGCCAACAATTTG3' and
5' GATCGCGATAATGCGATCCACC) for genomic DNA amplification and
5'GTATCTCGTAAATGCGCAG3' for sequencing.

*Fhos*$^{I966A}$ substitution allele: CRISPR-based genome editing was used to generate the point mutant allele *Fhos*$^{I966A}$ as described (*Gratz et al., 2013*). Briefly, three 1000 bp fragments, which cover the Fhos FH2 domain genomic region, including exon #20, were synthesized, and cloned into the pHD-DsRed-attB vector (Addgene). The two flanking fragments served as homology arms, while the middle fragment harbored a point mutation leading to substitution of Isoleucine 966 to Alanine. Two different gRNAs 5'CCTCATATAACACCCAATTGGTC and 5'CCATGTAAGAATTAACTTTTGTA) homologous to sites 5' and 3' of the replaced genomic area were synthesized and cloned into pU6-BbsI-chiRNA. The plasmid mixture was injected into *vasa*-Cas9 flies (BDSC#55821) (BestGene). All plasmid constructs were verified by sequencing.

## Fhos antibody production

Antibodies were raised to the 93 C-terminal residues of the Fhos protein. The relevant sequence was amplified by PCR from the SD08909 cDNA clone and cloned into the pDEST17-6xHis vector (Invitrogen). The recombinant protein expressed in BL21 cells, purified and injected into Wistar rats to raise the polyclonal antisera.

## Tissue preparation, immunostainings and confocal microscopy

A modified protocol from (*Weitkunat and Schnorrer, 2014*) was used for both adult and pupal IFM tissue preparation. Briefly, staged pupae were removed from the pupal case, pinned down on Sylgard plates and dissected in cold relaxation buffer (20 mM phosphate buffer, pH 7.0; 5 mM MgCl2; 5 mM EGTA, 5 mM ATP). For adult IFMs, thoraces of young adults (not older than 48 hr post enclosure) were bisected on the longitudinal axis and collected in the cold relaxation buffer. In both cases fixation was carried out with 4% paraformaldehyde for 20 (pupa) or 30 (adults) minutes at room

temp. Following washes and permabilization with PBS+0.3% Triton-X (pupa) and PBS+0.5% Triton-X (adults), the samples were incubated in blocking solution containing 0.1% bovine serum albumin (BSA) + 5% Normal goat serum (NGS). Staining that involved anti-α-actinin and/or anti-MHC required an additional 30 min blocking step with Image-IT FX signal enhancer reagent (Thermo) prior to the standard blocking step. All primary antibodies were diluted in standard blocking solution (0.1% BSA + 5% NGS) and were added for overnight incubation at 4°C. Following washes, secondary antibodies were added for 2 hr at room temp. Adult hemi-thoraces were cleared in 80% glycerol at 4°C overnight prior to mounting. All samples were mounted in Immu-Mount (Thermo).

Primary antibodies and dilutions used included: anti-GFP (chicken, 1:1000, Abcam); anti-MHC (rabbit, 1:1000, kindly provided by P.Fisher, Stony Brook); anti-α-actinin (rat, 1:50, Babraham institute, UK); anti-Zasp52 (rabbit 1:500, [*Katzemich et al., 2013*]); anti-Obscurin (rabbit, 1:500, [*Burkart et al., 2007*]); anti-Fhos (rat, 1:200) was generated as described above; anti-Tmod (Rat, 1:500, kindly provided by Velia Fowler).

Secondary antibodies used included Alexa Fluor 405, Alexa Flour 488, Alexa Fluor 555, Alexa Fluor 568 and Alexa Flour 647 conjugated to anti-rabbit, mouse, rat, or chick antibodies (Molecular Probes) and applied at a dilution of 1:1000. Atto647N-Phalloidin (Fluka) was used at 5 μg/ml.

Immunofluorescent images of fixed samples were acquired using Zeiss LSM 710 or Zeiss LSM 780 confocal scanning systems, equipped with a Zeiss Axiovert microscope, and using a ×20 0.8 N.A or ×63 oil immersion 1.4 N.A lenses. The initial image acquisition was performed using the imaging system Zen software.

## Transmission electron microscopy

Thoraces of young adults or isolated IFMs were collected in the ice cold relaxation buffer (20 mM phosphate buffer, pH 7.0; 5 mM MgCl2; 5 mM EGTA, 5 mM ATP). Following 15 min incubation the samples were transferred into 1 mM sodium cacodylate buffer (pH 7.4) containing fixative (4% para-formaldehyde and 2.5% glutaraldehyde). Samples were fixed for 1 hr (IFMs) or 2 hr (adults) at room temp and transferred to 4°C overnight. Samples were washed x3 with sodium cacodylate buffer, post fixed in 1% $OSO_4$ solution for 1 hr at room temp, washed x3 with sodium cacodylate buffer, incubated in 2% aqueous uranyl acetate for 1 hr and washed x3 with distilled water. Samples were taken through an ethanol dehydration series and incubated in propylene oxide (x3, 10 min each). Infiltration was performed with a series of propylene oxide: Epon mixtures, culminating in incubation in 100% Epon (x3, 12 hr each). Infiltrated samples were embedded in plastic moulds (EMS) and polymerized for 48 hr at 60°. Ultra thin sections were cut using diamond knife 35° (Diatome, Switzerland) on a Leica Reichert ultra cut UCT. Sections were post stained with 1% lead citrate and 2% uranyl acetate.

Images were recorded using an FEI T12 spirit BioTWIN transmission electron microscope (TEM) operating at 120KV and equipped with an Eagle 2Kx2K CCD camera (FEI).

## Data analysis

Measurements of various geometric properties of the sarcomere and tagged actin monomers incorporation were performed using Fiji image analysis software. For sarcomere length and width, the Fiji measurement tool was used to draw a vertical line (width) or horizontal line (length) across the Z-disc of a single sarcomere from pointed end to pointed end, using phalloidin staining as a guide. 50 sarcomeres were measured form 7 different flies for each genotype (350 sarcomeres in total). Horizontal lines or polygons were drawn to measure the Z-disc associated incorporation band length (act5C control and following *Fhos* knockdown) or the incorporation frame area (act88F) respectively. 50 sarcomeres were measured form 4 different flies for each genotype (200 sarcomeres in total). The intensity distribution in *Figure 4E and G* was measured along 10 μm horizontal profiles starting from a Z-disc and drawn at the middle of the myofibril. The data represent an average of normalized values collected from 15 profiles in 7 different flies (n = 105) for each genotype. Vertical profiles drawn across half sarcomeres was used to measure the GFP intensity distribution in *Figure 4J*. The data represents an average of normalized values collected from 100 half sarcomeres in 4 different flies (400 sarcomeres in total). To measure the M/Z incorporation intensity ratio in *Figure 5 and S5* polygons were drawn around the Z-disc or M-line area. The date represents an intensity measurement from 50 sarcomeres in 5 different flies (250 sarcomeres in total). The F-actin intensity distribution in

*Figure 5 and S5* was measured along a 13 µm horizontal profile. The data represents an average of normalized values collected from 15 profiles in 5 different flies (n = 75). Distribution of actin incorporation events was obtained from 50 sarcomeres in 5 different flies (250 sarcomeres in total). The incorporation events were visualized by a GFP intensity profile drawn along the contour of the nascent arrays. Z-disc and M-line vicinity markers (Zasp52 and Obscurin, respectively) determined the location of the incorporation event.

For counting thick filaments in TEM cross-sections, a threshold base segmentation was applied and the number of filaments determined by using Fiji Analyze Particles tool. The data represent an average thick filament number per myofibril from 20 myofibrils from 3 different samples.

All graphs and statistic tests were done using GraphPad Prism software. The figures were assembled and organized using Adobe Photoshop CS6.

## Acknowledgements

We wish to thank our lab manager Shari Carmon and BestGene (Chino Hills, CA) for generating the CRISPR-based short isoform mutants and I966A mutant flies, Gali Housman for help in assessing formin knockdown phenotypes and the WIS Electron Microscopy and Antibody Units for technical support. Reagents were kindly provided by Sven Bogdan (U. Munster), Belinda Bullard (U. of York), Paul Fisher (SUNY Stony Brook), Velia Fowler (Scripps Research Institute), Frieder Schöck (McGill U.), the Babraham Institute (Cambridge), the Drosophila Genomics Resource Center (Indiana U.), the Vienna Drosophila Research Center, the TRiP stock center (Harvard Med. School) and the Bloomington Drosophila Stock Center (Indiana U.). We thank our colleagues in the Shilo lab for their continuous input and encouragement. The authors declare no competing financial interests. This work was supported by grants from the Israel Science Foundation and the Weizmann-UK Joint Research Program to EDS and B-ZS. B-ZS is an incumbent of the Hilda and Cecil Lewis chair in Molecular Genetics.

## Additional information

### Funding

| Funder | Grant reference number | Author |
| --- | --- | --- |
| Israel Science Foundation | 557/15 | Eyal D Schejter<br>Ben-Zion Shilo |
| Weizmann UK | Joint research program | Eyal D Schejter<br>Ben-Zion Shilo |

The funders had no role in study design, data collection and interpretation, or the decision to submit the work for publication.

### Author contributions

AS, Conception and design, Acquisition of data, Analysis and interpretation of data, Drafting or revising the article; ND, Acquisition of data, Analysis and interpretation of data; EDS, B-ZS, Conception and design, Analysis and interpretation of data, Drafting or revising the article

### Author ORCIDs

Ben-Zion Shilo, http://orcid.org/0000-0003-4903-8889

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
