## [Decision Letter]

[Editors’ note: this article was originally rejected after discussions between the reviewers, but the authors were invited to resubmit after an appeal against the decision.]

Thank you for submitting your work entitled "The *Drosophila* formin Fhos is a primary mediator of sarcomeric thin-filament array assembly" for consideration by *eLife*. Your article has been reviewed by three peer reviewers (with expertise in *Drosophila* cell biology, cytoskeleton, and muscle cell biology), and the evaluation has been overseen by a Reviewing Editor and Vivek Malhotra as the Senior Editor. The following individuals involved in review of your submission have agreed to reveal their identity: Velia Fowler (Reviewer #2).

Our decision has been reached after consultation between the reviewers. Based on these discussions and the individual reviews below, we regret to inform you that your work will not be considered further for publication in *eLife*.

In discussions, the referees agreed that the description of the stages involved in assembly of thin filaments was very thorough and well done. The referees agreed that the quality of data was very high as well.

However, several questions were raised (please see referee comments verbatim), in particular on the extent of mechanistic advance and the generality of the conclusions, considering previous work on other formins in muscle and the sals.

Reviewer #1:

The organization of actin-based filaments is central to function of sarcomere, the basic functional units of muscle fibres. In *Drosophila*, indirect flight musculature (IFM) is a well-established skeletal muscle model system for studying assembly and maturation of thin-filament arrays. This study (Arkadi Shwartz et al.) described the role of Fhos, the single *Drosophila* homolog of the FHOD sub-family of formins, as a primary mediator of indirect flight musculature (IFM) thin filament organization. The authors also identified several phases in the dynamic construction of thin-filament arrays: intensive microfilament synthesis -> assembly of nascent arrays -> elongation primarily from filament pointed-ends -> radial growth of the arrays via recruitment of peripheral filaments -> continuous barbed-end turnover. In addition, the authors showed that the WH2-domain protein Sals that is known to contribute to pointed-end filament elongation, was specifically responsible for pointed-end elongation.

Unfortunately, this manuscript lacks conceptual novelty and is not suitable for general readers of *ELife*. Other formin proteins such as Diaphanous has already been studied in skeletal muscle model and this manuscript on Fhos, another formin, does not provide significant mechanistic insight into actin-based thin filaments formation in sarcomere. Given that Sals was previously known to contribute to pointed-send filament elongation, the data on Sals in this manuscript also lacks novelty.

Reviewer #2:

This is an interesting study of the role of *Drosophila* Fhod subfamily formins (Fhos) in thin filament assembly into myofibrils in the *Drosophila* indirect flight muscle (IFM). The IFM is a great model for sarcomere and thin filament assembly, and the authors take advantage of the fly genetics as well as the ability to express 'pulses' of GFP-actin during IFM myofibril assembly to evaluate the role of the Fhos proteins in actin incorporation/assembly at the barbed vs pointed ends of the thin filaments, as well as assembly of new peripheral thin filaments during stages of myofibril growth. This is an important area in which relatively few careful studies combining genetic deletion or knockdown with tests of actin incorporation have been undertaken. The key findings are that the long isoform of Fhos (Fhos-PH) is required for actin assembly at barbed ends and peripheral thin filament addition during myofibril growth, and that the nucleation activity of Fhos is not essential for this function. In addition, data is presented suggesting that the SALS protein is important for pointed end actin addition and thin filament elongation. The data is of high quality and interesting and reinforce and extend substantially the earlier findings from Mardahl-Dumesnil and Fowler (2001) that thin filament elongation during myofibril assembly takes place at their pointed but not barbed ends, and from Bai et al. (2007) that SALS is important for thin filament elongation at pointed ends. There are a few areas that require clarification and extension to solidify the conclusions.

1) The image of the thin myofibrils in Figure 3 showing partial rescue of the Fhos MI01421 with the GFP-Fhos-PA (short Fhos) is described as having "only weak corrective influence on the disrupted microfilament organization and other sarcomeric defects in myofibrils (Figure 3)". However, to my eye this image shows thin myofibrils but with clearly demarcated and short sarcomeres, quite similar to the image of the myofibrils for the point mutant Fhos I966A in Figure 6', which is described as having "an organized structure of repeated sarcomeric units, with clear demarcation of the Z-discs (Figure 6'), suggesting an arrest in sarcomere growth following proper initial assembly and organization." Thus, it appears to me that the rescue with the short GFP-Fhos-PA should be reinterpreted, and the text rewritten. See next point also.

2) The experiment in Figure 4 shows that the Fhos-i knockdown (which reduces long Fhos isoforms specifically) reduces GFP-actin incorporation at Z lines and at the myofibril periphery, but does not affect the GFP-actin incorporation at the M line. This is a very nice experiment. However, since the short Fhos-PA localizes to the M lines (Figure 3'), and the Fhos-PA does rescue the myofibril disruption phenotype to some extent (Figure 3), this could mean that the short Fhos-PA functions at M lines to promote GFP-actin incorporation. However, this was not tested. Depending on the results, the model in Figure 7 may need to be modified.

3) The GFP-actin (act88F) incorporation in the sals RNAi experiment (Figure 5') is not convincingly qualitatively different from the control in Figure 5'. I can see both M and Z line incorporation, just that some M lines appear fainter than others, and overall the image appears to have reduced/fuzzier GFP-actin incorporation. (Note that the shorter sarcomeres due to reduction of sals are convincing (Figure 2—figure supplement 2) and repeat the previous study from Bai et al.) However, I wonder whether the sals is actually affecting pointed end actin elongation directly. Do the authors have stronger data for the GFP-actin incorporation locations? Can the relative M and Z line incorporation be quantified by line scans to compare the WT and the knockdown?

4) Along these same lines, the study of Mardahl-Dumesnil and Fowler implied that Tmod reduces actin pointed end incorporation and thus prevents elongation. However, these authors never tested a Tmod knockdown or deletion, or actin incorporation. Can the authors perform a knockdown of fly Tmod in the IFM to determine whether this affects GFP-actin incorporation at the pointed end? This would fill in the picture, and strengthen the unique role(s) of the Fhos isoforms.

5) In Figure 6, the overall thin/thick filament organization appears normal by TEM of cross sections in the small myofibrils that form in the Fhos I966A point mutant (Figure 6). However, I noticed a few areas of discontinuity in the lattice (upper left of myofibril) in which there are missing or extra thin filament surrounding the thick filament. In Figure 6, there is an impression that the M line is not perfectly straight and that H zone is not perfectly even, so that some thin filaments may be shorter and some longer. I wonder whether there may be some subtle thin filament length or organization defects. Can the authors comment on this?

6) In Figure 1, can higher magnification images of the GFP-actin88F for the 45 h incorporation be shown to help identify the location of the actin "patches"? It appears that they do not colocalize with the Z line marker, Zasp. Do they colocalize with the pointed end marker, Tmod? Similarly, in the FhosMI-GFP experiment in Figure 3', where are the FhosMI-GFP patches? The colocalization with α-actinin is not shown at high magnification. (The regular sarcomeres at later times make it easier to figure out what is going on).

Reviewer #3:

The study describes 3 stages of actin monomer incorporation during *Drosophila* indirect flight muscle differentiation. Initial bulk polymerisation is followed by ordering into units, which are then elongated and widened to match the uniform sarcomere size. The formin FHOD is implicated in organising and thickening the sarcomere, which requires its long isoforms with N-terminal Z-disk localisation domain. Sals is required for pointed end elongation (as had been shown previously). In both these areas actin88F is incorporated, while at the barbed end, a different actin isoform, actin5c is being built in in a FHOD independent manner.

FHOD seems not to require its barbed-end association (and any associated nucleation, elongation and capping function) until about 60h APF as a conserved mutation I966A can still form organised sarcomeres. Thus the authors argue that FHOD might use its bundling activity during the early stages. However, such a bundling activity is neither shown for FHOD in the present study, nor in the two cited papers.

It is slightly puzzling why FHOD needs to localise to the barbed end, but doesn't need to actually bind to the barbed end. Any mechanistic explanations provided are speculative as no data as to the activity of the mutant or mechanistic action of FHOD or Sals itself are contained in the paper. Thus interpretations rest on the assumption that these would function as other formins. Interestingly, FHOD3, the closest mammalian homologue of FHOD inhibits barbed end elongation, rather than accelerating it (Taniguchi 2009) – a fact that seems not to be discussed in this study.

While the N-terminal extension of the longer FHOD isoforms seem to be of functional importance to organise sarcomeres and the longer isoforms localise specifically to the Z-disk, a causal relationship cannot be drawn, i.e. it is not clear that the localisation itself is required or that the localisation is sufficient to enable the canonical elements of FHOD to fully function. This would need to be tested with more subtle mutations that affect localisation (rather than the removal of 120kD!) and/or using means to bypass the requirement for the N-terminal domain by a more direct recruitment of the shorter FHOD isoforms to Z-disks. In the absence of such mechanistic experiments, the observations are difficult to interpret and some of the statements would need to be altered or stated as being purely speculative.

To my understanding, no data show that radial size increase occurs through recruitment of peripheral filaments – instead incorporation of actin monomers suggest their new synthesis. Also, the barbed end-binding activity of FHOD seems to be required for this step, whether nucleating or inhibitory.

Overall a beautifully illustrated descriptive story identifying temporal and spatial patterns of actin incorporation and formin activity. However, the authors have over-interpreted their findings and make mechanistic claims that are not substantiated by data in the study (or elsewhere).

[Editors’ note: what now follows is the decision letter after the authors submitted for further consideration.]

Thank you for submitting your article "The *Drosophila* formin Fhos is a primary mediator of sarcomeric thin-filament array assembly" for consideration by *eLife*.

Your appeal has been reviewed by an additional peer reviewer, and the evaluation has been overseen by a Reviewing Editor and Vivek Malhotra as the Senior Editor. The reviewer has opted to remain anonymous.

While the additional referee sees the value of your description of actin polymerization and thin filament assembly in *Drosophila* IFM, they have raised a number of points, all of which seem very reasonable and geared towards tightening controls and improving imaging quality.

In summary, we would like you to revise the manuscript, taking into consideration the comments of all the four referees, and in particular on the substantive points raised by referees 2, 3, and 4. You already have the comments of reviewer# 1, 2 and 3, and below are the comments of reviewer #4

Reviewer #4:

Arkadi Shwartz et al. present a study on *Drosophila* indirect flight muscle actin assembly during the various stages of development and observe the incorporation of different actin isoforms (act88F, act5c) into specific locations of the sarcomere. The authors find that Fhos the single *Drosophila* homologue of the mammalian FHOD subfamily, which has a documented role in sarcomere assembly and/or maturation, is crucially required for IFM integrity. Knockdown and a previously described mutation lead to thin and randomly oriented myofibrils, with defects appearing after 50hs APF, suggesting an involvement at early stages of sarcomere formation. They further analyze the involvement of different isoforms as well as SALS in the regulation of actin assembly and find that the long isoform is targeted to the Z-disc, where it assembles and bundles the actin, while the short isoform is localized to the pointed ends and dispensable for myofibril formation. SALS, in contrast, as previously demonstrated (Bai et al., 2007) is needed to regulate the thin filament length at the pointed end. The article provides an unprecedented level of detail regarding the different stages of actin assembly during myofibrillogenesis and the involvement of FHOD proteins therein. There are however several points that need to be addressed before publication.

Major points:

1) The data presented regarding the unsuccessful rescue of the Fhos deletion phenotype by FHOS-PA is not entirely convincing. While there is an apparent lack of rescue in Figure 3—figure supplement 1', the stronger overexpression of Fhos-PA results in clearly thicker and more organized sarcomeres in Figure 3, suggesting that the short isoform can, at least at high expression levels take over the function and even localize to the Z-disc (Figure 3—figure supplement 1'). The data from targeting the 5'coding exon provides further evidence for the involvement of the large isoforms (Figure 3'), however, the authors should perform rescue experiments with the PH isform as well, to show that it can fully restore the myofibril integrity.

2) Further, regarding the isoforms and the conclusions that the authors draw from their data:

a) The authors suggest that PA and PH are the main isoforms, but no data (qPCR or similar) is presented, or studies cited, to this regards.

b) Disregarding the expression levels, a rescue with the π isoform and localization of the π isoform should be tested as well to exclude major involvement of this isoform.

3) The authors state that the short isoform is inactive, due to its localization to the vicinity of the M-band (3H,H'). This is however not supported by the presented data since the authors find 1) addition of actin to M-Bands, and thickening of myofibrils at late stages of development (Figure 1); 2) the authors argue, referencing the data from Schoenichen et al. for FHOD1, that Fhos has an important actin bundling activity, which requires filament side binding, however not necessary barbed end binding. 3) stage specific localization changes have been reported for FHOD3 as well, with early localization to the vicinity of the M-Band (Iskratsch et al., 2013). 4) the data from Figure 4 demonstrates a patchy incorporation of actin at 60hrs, not only at the Z-disc, which is lost after knockdown of Fhos. This demonstrates that localization to the Z-disc is not necessarily the single determinant of actin assembly activity actin filament turnover could be contributing to differing localization, especially since it was shown in living zebrafish that YFP-actin is incorporated all along the I-Band(Sanger et al., 2009). The authors could test this by performing FRAP experiments to investigate Actin turnover at the Z-disc, around the M-Band and the periphery of the sarcomeres in presence or absence of the various isoforms.

4) Several of the images comparing the control and Fhos deletion/knock downs are difficult to compare due to varying quality, with fuzzier staining also present in the other channels (especially F-actin and zasp in Figure 4 vs I and J vs K, and F-actin vs Obscurin Figure 5 vs E). The authors should provide better comparable images (or if not possible, discuss the reason for this difference). Also the authors should provide quantifications of the intensities (e.g. of the M and Z-band intensities normalized to the background).

5) The GFP-act88F localization around the M-bands is narrower in the sals-i, but still present. The authors should provide quantifications for the width (i.e. does this account for the difference in thin filament length) and test or at least discuss what other proteins could be responsible for the ongoing pointed end assembly in absence of sals.

---

## [Author Response]

[Editors’ note: the author responses to the first round of peer review follow.]

*Reviewer #1:*

*The organization of actin-based filaments is central to function of sarcomere, the basic functional units of muscle fibres. In Drosophila, indirect flight musculature (IFM) is a well-established skeletal muscle model system for studying assembly and maturation of thin-filament arrays. This study (Arkadi Shwartz et al.) described the role of Fhos, the single Drosophila homolog of the FHOD sub-family of formins, as a primary mediator of indirect flight musculature (IFM) thin filament organization. The authors also identified several phases in the dynamic construction of thin-filament arrays: intensive microfilament synthesis -> assembly of nascent arrays -> elongation primarily from filament pointed-ends -> radial growth of the arrays via recruitment of peripheral filaments -> continuous barbed-end turnover. In addition, the authors showed that the WH2-domain protein Sals that is known to contribute to pointed-end filament elongation, was specifically responsible for pointed-end elongation.*

*Unfortunately, this manuscript lacks conceptual novelty and is not suitable for general readers of ELife. Other formin proteins such as Diaphanous has already been studied in skeletal muscle model and this manuscript on Fhos, another formin, does not provide significant mechanistic insight into actin-based thin filaments formation in sarcomere. Given that Sals was previously known to contribute to pointed-send filament elongation, the data on Sals in this manuscript also lacks novelty.*

We take issue with the reviewer’s assessment of the novelty of our findings vis-a-vis published studies on Formin function during skeletal muscle sarcomerogenesis. Our study couples a detailed analysis of the dynamics of thin-filament array assembly, at high spatial and temporal resolution, with assignment of distinct functional roles to a single Formin sub-family (FHOD/Fhos). We are unaware of similar comprehensive studies, or of such clear demonstrations of sarcomeric Formin function, and believe that we have provided considerable novel insight, using state-of-the-art tools, to a fundamental and somewhat neglected aspect of skeletal muscle biology. Our analysis of Sals function complements the assignment of functional roles to Fhos, and generalizes the significance of this element to sarcomere formation.

*Reviewer #2:*

*This is an interesting study of the role of Drosophila Fhod subfamily formins (Fhos) in thin filament assembly into myofibrils in the Drosophila indirect flight muscle (IFM). The IFM is a great model for sarcomere and thin filament assembly, and the authors take advantage of the fly genetics as well as the ability to express 'pulses' of GFP-actin during IFM myofibril assembly to evaluate the role of the Fhos proteins in actin incorporation/assembly at the barbed vs pointed ends of the thin filaments, as well as assembly of new peripheral thin filaments during stages of myofibril growth. This is an important area in which relatively few careful studies combining genetic deletion or knockdown with tests of actin incorporation have been undertaken. The key findings are that the long isoform of Fhos (Fhos-PH) is required for actin assembly at barbed ends and peripheral thin filament addition during myofibril growth, and that the nucleation activity of Fhos is not essential for this function. In addition, data is presented suggesting that the SALS protein is important for pointed end actin addition and thin filament elongation. The data is of high quality and interesting and reinforce and extend substantially the earlier findings from Mardahl-Dumesnil and Fowler (2001) that thin filament elongation during myofibril assembly takes place at their pointed but not barbed ends, and from Bai et al. (2007) that SALS is important for thin filament elongation at pointed ends. There are a few areas that require clarification and extension to solidify the conclusions.*

*1) The image of the thin myofibrils in Figure 3 showing partial rescue of the Fhos MI01421 with the GFP-Fhos-PA (short Fhos) is described as having "only weak corrective influence on the disrupted microfilament organization and other sarcomeric defects in myofibrils (Figure 3)". However, to my eye this image shows thin myofibrils but with clearly demarcated and short sarcomeres, quite similar to the image of the myofibrils for the point mutant Fhos I966A in Figure 6', which is described as having "an organized structure of repeated sarcomeric units, with clear demarcation of the Z-discs (Figure 6'), suggesting an arrest in sarcomere growth following proper initial assembly and organization." Thus, it appears to me that the rescue with the short GFP-Fhos-PA should be reinterpreted, and the text rewritten. See next point also.*

These points are well taken, and we have now added new observations and have substantially revised our presentation of rescue and localization data, and our discussion of their significance. A key new experiment is specific disruption of the short Fhos isoforms (via CRISPR-based indels- Figure 3 and Figure 3—figure supplement 1). Using this approach we can now state with confidence that the short isoforms are dispensable and the long isoforms are sufficient for full sarcomeric function. The inability of the short forms to provide significant function when expressed at physiological levels is demonstrated both by the severe phenotypes resulting from specifically disrupting the long isoforms, and by the lack of rescue of null alleles by UAS-Fhos-PA, when driven by *armadillo*-GAL4. We now show this result in figure panels 3D-D’, as was our original intention. We fully agree with the reviewer that over-expression of the short form results in partial rescue- as we show (Figure 3—figure supplement 1”), such rescue is likely the result of “ectopic” localization of the short form to the Z-disc, and, in fact, corresponds to our view that Z-disc localization is critical for Fhos function.

*2) The experiment in Figure 4 shows that the Fhos-i knockdown (which reduces long Fhos isoforms specifically) reduces GFP-actin incorporation at Z lines and at the myofibril periphery, but does not affect the GFP-actin incorporation at the M line. This is a very nice experiment. However, since the short Fhos-PA localizes to the M lines (Figure 3'), and the Fhos-PA does rescue the myofibril disruption phenotype to some extent (Figure 3), this could mean that the short Fhos-PA functions at M lines to promote GFP-actin incorporation. However, this was not tested. Depending on the results, the model in Figure 7 may need to be modified.*

The Fhos-i knockdown in this figure was based on an siRNA construct targeting ALL isoforms. In any case, we now find (upon specific elimination of the short forms) that the fully functional (endogenous) long isoforms localize to both the Z-disc and M-line regions, while Fhos-PH, an isoform that localizes strictly to the Z-disc and contains only part of the large N-terminal region, provides only partial rescue of null alleles. All this leads us to conclude that Fhos may well function at sites other than the Z-disc (where it is essential), including the M-line. While the nature of these non-Z disc roles is currently unknown, we have modified the text to reflect this notion.

*3) The GFP-actin (act88F) incorporation in the sals RNAi experiment (Figure 5') is not convincingly qualitatively different from the control in Figure 5'. I can see both M and Z line incorporation, just that some M lines appear fainter than others, and overall the image appears to have reduced/fuzzier GFP-actin incorporation. (Note that the shorter sarcomeres due to reduction of sals are convincing (Figure 2—figure supplement 2) and repeat the previous study from Bai et al.) However, I wonder whether the sals is actually affecting pointed end actin elongation directly. Do the authors have stronger data for the GFP-actin incorporation locations? Can the relative M and Z line incorporation be quantified by line scans to compare the WT and the knockdown?*

We now use fluorescence intensity measurements to provide quantification of both the sarcomere shortening (Figure 5) and reduced M/Z incorporation ratio (Figure 5) phenotypes associated with *sals* knockdown.

*4) Along these same lines, the study of Mardahl-Dumesnil and Fowler implied that Tmod reduces actin pointed end incorporation and thus prevents elongation. However, these authors never tested a Tmod knockdown or deletion, or actin incorporation. Can the authors perform a knockdown of fly Tmod in the IFM to determine whether this affects GFP-actin incorporation at the pointed end? This would fill in the picture, and strengthen the unique role(s) of the Fhos isoforms.*

Tmod is indeed a logical candidate for an element involved in pointed end elongation. As recommended by the reviewer, we assessed and analyzed the effects of knockdown in similar fashion to the study of *sals*. In general the *tmod* knockdown phenotypes are quite mild, and possibly indicate an early role for Tmod in thin-filament array assembly. The relevant observations are shown in Figure S5.

*5) In Figure 6, the overall thin/thick filament organization appears normal by TEM of cross sections in the small myofibrils that form in the Fhos I966A point mutant (Figure 6). However, I noticed a few areas of discontinuity in the lattice (upper left of myofibril) in which there are missing or extra thin filament surrounding the thick filament. In Figure 6, there is an impression that the M line is not perfectly straight and that H zone is not perfectly even, so that some thin filaments may be shorter and some longer. I wonder whether there may be some subtle thin filament length or organization defects. Can the authors comment on this?*

We thank the reviewer for pointing out these imperfections in the Fhos I966A mutant arrays. However, given that these were obtained from 1 day old adult flies, following several days of IFM growth arrest, they may well be the result of secondary effects. We therefore prefer not to speculate about their significance, and to limit our interpretation to the general resemblance they bear to intermediate stage IFMs.

*6) In Figure 1, can higher magnification images of the GFP-actin88F for the 45 h incorporation be shown to help identify the location of the actin "patches"? It appears that they do not colocalize with the Z line marker, Zasp. Do they colocalize with the pointed end marker, Tmod? Similarly, in the FhosMI-GFP experiment in Figure 3', where are the FhosMI-GFP patches? The colocalization with α-actinin is not shown at high magnification. (The regular sarcomeres at later times make it easier to figure out what is going on).*

Following these comments, we have extended our analysis of the actin “patches”, by adding magnified panels and fluorescence intensity profiles for 45 hr IFMs stained for GFP-actin, phalloidin and the Z-disc and M-like markers Zasp and Obscurin (Figure 1—figure supplement 1). Further quantification clearly demonstrates that the patches tend to localize to the ends of the arrays (Figure 1—figure supplement 1).

*Reviewer #3:*

*The study describes 3 stages of actin monomer incorporation during Drosophila indirect flight muscle differentiation. Initial bulk polymerisation is followed by ordering into units, which are then elongated and widened to match the uniform sarcomere size. The formin FHOD is implicated in organising and thickening the sarcomere, which requires its long isoforms with N-terminal Z-disk localisation domain. Sals is required for pointed end elongation (as had been shown previously). In both these areas actin88F is incorporated, while at the barbed end, a different actin isoform, actin5c is being built in in a FHOD independent manner.*

FHOD seems not to require its barbed-end association (and any associated nucleation, elongation and capping function) until about 60h APF as a conserved mutation I966A can still form organised sarcomeres. Thus the authors argue that FHOD might use its bundling activity during the early stages. However, such a bundling activity is neither shown for FHOD in the present study, nor in the two cited papers.

We feel that our study covers considerable ground in breaking down the process of thin-filament array assembly into a discrete series of continuous events, some of which overlap temporally, and in assigning functional roles to several key elements that mediate these processes. Admittedly, we do not provide biochemical data that tests the actin-associated activities of Fhos (not for lack of trying- Fhos expression at useful levels has proven to be very difficult). However, we feel compelled to use the ample descriptive and genetic data provided to speculate on molecular mechanisms, bearing in mind established molecular capacities of the proteins involved. That said, we respect the reviewer’s expert assessment that some of the statements come across as over-interpretation of the data, and have tried to reorganize the text and “tone down” the language (in the main text and the relevant mention in the abstract) accordingly.

With regard to FHOD formins acting as microfilament bundling proteins- here we feel fully justified in building upon this notion, which was clearly demonstrated by Schonichen et al. (JCS 2013) using a number of assays, and is treated as an established capacity of FHOD proteins in various papers including the study from Kutscheidt et al. (NCB 2014) which we cite. However, we agree that initial mention of this specific activity was granted excessive prominence, and has now been moved from the Results section to the Discussion. Further mention of a bundling capacity and its significance are now phrased in a less dogmatic fashion.

*It is slightly puzzling why FHOD needs to localise to the barbed end, but doesn't need to actually bind to the barbed end. Any mechanistic explanations provided are speculative as no data as to the activity of the mutant or mechanistic action of FHOD or Sals itself are contained in the paper. Thus interpretations rest on the assumption that these would function as other formins. Interestingly, FHOD3, the closest mammalian homologue of FHOD inhibits barbed end elongation, rather than accelerating it (Taniguchi 2009) – a fact that seems not to be discussed in this study.*

We fully agree that the critical nature of Fhos localization to the Z-disc region does not readily correspond with its presumed molecular capacities. As brought up in the Discussion, Z-disc region localization appears to be a general feature of FHOD proteins in striated muscle, and resolving this matter is a priority for future studies on sarcomeric FHOD/Fhos function. Clearly, determination of the microfilament-regulating properties of Fhos should be a central aspect of such investigations. Our data only allows us to speculate regarding possible mechanisms, but we feel compelled to do so. Again, we have revised the tone of our discussion to properly reflect its speculative nature. The Taniguchi et al. study is now cited as a reference for the “poor nucleation” capacity of FHOD-family proteins.

*While the N-terminal extension of the longer FHOD isoforms seem to be of functional importance to organise sarcomeres and the longer isoforms localise specifically to the Z-disk, a causal relationship cannot be drawn, i.e. it is not clear that the localisation itself is required or that the localisation is sufficient to enable the canonical elements of FHOD to fully function. This would need to be tested with more subtle mutations that affect localisation (rather than the removal of 120kD!) and/or using means to bypass the requirement for the N-terminal domain by a more direct recruitment of the shorter FHOD isoforms to Z-disks. In the absence of such mechanistic experiments, the observations are difficult to interpret and some of the statements would need to be altered or stated as being purely speculative.*

We think that we have provided a considerable body of data supporting the notion that Z-disk localization is critical nature:

a) The active long isoforms- which are absolutely necessary for sarcomeric Fhos fuction- localize to this site;

b) The short forms do not localize there under physiological conditions, and completely fail to provide function (although they are fully sufficient for Fhos fuction in all other tissues);

c) Partial rescue by the short form is provided by its ectopic localization to the Z-disc region, under circumstances of over-expression.

We believe that this set of results justifies our conclusions regarding the significance of Z-disc localization. We agree that identification and manipulation of a minimal localization sequence within the long isoform is warranted, and would serve as an important step in refining and understanding the nature of Fhos localization within the sarcomere, but this must be left to future studies.

*To my understanding, no data show that radial size increase occurs through recruitment of peripheral filaments – instead incorporation of actin monomers suggest their new synthesis. Also, the barbed end-binding activity of FHOD seems to be required for this step, whether nucleating or inhibitory.*

We are in agreement with the reviewer on this point- namely, that newly-synthesized microfilaments constitute a major portion (all?) of the filaments added to the periphery of the sarcomere “core”, and stated as such in the Discussion text. We agree and similarly state that barbed-end binding by Fhos is required for its involvement in radial growth of the array. The question we try to grapple with is why a FHOD-type formin, unlikely to be responsible for synthesis, is still essential for radial growth. Bundling is suggested as a possible mechanism.

*Overall a beautifully illustrated descriptive story identifying temporal and spatial patterns of actin incorporation and formin activity. However, the authors have over-interpreted their findings and make mechanistic claims that are not substantiated by data in the study (or elsewhere).*

We thank the reviewer both for the positive assessment of our experimental work and for pointing out the shortcomings of our interpretation, which we have tried to revise accordingly.

[Editors’ note: the author responses to the re-review follow.]

*The additional referee has kindly provided a detailed critique within a few days. While this referee sees the value of your description of actin polymerization and thin filament assembly in Drosophila IFM, the referee has raised a number of points, all of which seem very reasonable and geared towards tightening controls and improving imaging quality.*

*Reviewer #4:*

*Major points:*

*1) The data presented regarding the unsuccessful rescue of the Fhos deletion phenotype by FHOS-PA is not entirely convincing. While there is an apparent lack of rescue in Figure 3—figure supplement 1', the stronger overexpression of Fhos-PA results in clearly thicker and more organized sarcomeres in Figure 3, suggesting that the short isoform can, at least at high expression levels take over the function and even localize to the Z-disc (Figure 3—figure supplement 1'). The data from targeting the 5'coding exon provides further evidence for the involvement of the large isoforms (Figure 3'), however, the authors should perform rescue experiments with the PH isform as well, to show that it can fully restore the myofibril integrity.*

These are all good points, some of which were raised by Reviewer #2 as well (please also see our answers there). With regard to the function of the short isoform- we agree, and now better present the data, which we believe to show that rescue by the short forms is possible, but requires localization to the Z-disc. As for the rescuing capacity of the long forms- the CRISPR-based mutation we have now generated, specifically disrupting the short forms, demonstrates that the long forms are both necessary (as shown before) and sufficient. Rescue by the PH form (Figure 3—figure supplement 1’’’) is partial, but while this construct localizes strictly to the Z-disc region, rescue by the endogenous long isoforms is associated with localization to both the Z-disc and M-line regions (Figure 3—figure supplement 1). We conclude and now state that while Z-disc localization is critical, M-line localization may be of functional importance as well.

2) Further, regarding the isoforms and the conclusions that the authors draw from their data:

*a) The authors suggest that PA and PH are the main isoforms, but no data (qPCR or similar) is presented, or studies cited, to this regards.*

The use of the term “main” is in error- it has been changed to “representative”, and we thank the reviewer for pointing it out. Form PA is identical in coding sequence to most other short isoforms, and was used by Bogdan and colleagues in their study of Fhos function, to which we refer throughout. PH was used as a proxy for the long forms, since a corresponding cDNA was available to generate reagents.

*b) Disregarding the expression levels, a rescue with the π isoform and localization of the π isoform should be tested as well to exclude major involvement of this isoform.*

Our new results now suggest, in fact, that the longer forms (PI and PJ) are the ones to provide full functionality, but unfortunately, we were not successful in attempts to generate matching constructs.

*3) The authors state that the short isoform is inactive, due to its localization to the vicinity of the M-band (3H,H'). This is however not supported by the presented data since the authors find 1) addition of actin to M-Bands, and thickening of myofibrils at late stages of development (Figure 1); 2) the authors argue, referencing the data from Schoenichen et al. for FHOD1, that Fhos has an important actin bundling activity, which requires filament side binding, however not necessary barbed end binding. 3) stage specific localization changes have been reported for FHOD3 as well, with early localization to the vicinity of the M-Band (Iskratsch et al., 2013). 4) the data from Figure 4 demonstrates a patchy incorporation of actin at 60hr, not only at the Z-disc, which is lost after knockdown of Fhos. This demonstrates that localization to the Z-disc is not necessarily the single determinant of actin assembly activity actin filament turnover could be contributing to differing localization, especially since it was shown in living zebrafish that YFP-actin is incorporated all along the I-Band(Sanger et al., 2009). The authors could test this by performing FRAP experiments to investigate Actin turnover at the Z-disc, around the M-Band and the periphery of the sarcomeres in presence or absence of the various isoforms.*

As noted above, our genetic data now supports the notion that Z-disc/barbed-end associated function is not the entire story. While we do not have solid ideas regarding M-line function of Fhos (pointed-end elongation appears to be Fhos-independent), we stress this additional layer of functional complexity in the text.

*4) Several of the images comparing the control and Fhos deletion/knock downs are difficult to compare due to varying quality, with fuzzier staining also present in the other channels (especially F-actin and zasp in Figure 4 vs I and J vs K, and F-actin vs Obscurin Figure 5 vs E). The authors should provide better comparable images (or if not possible, discuss the reason for this difference). Also the authors should provide quantifications of the intensities (e.g. of the M and Z-band intensities normalized to the background).*

We have now improved data presentation according to the reviewer’s comments:

a) In Figure 4’, I’ and J we quantified the distribution of actin intensity across the arrays, demonstrating enhanced wildtype incorporation at the periphery, which is gone following *Fhos* knockdown.

b) In Figure 4 we quantified the width of the incorporation band at the Z-disc, an analysis that demonstrates that this turnover is Fhos independent

In this context we wish to note that- as the reviewer observed- the GFP incorporation signal is generally “fuzzier” reviewer upon *fhos* knockdown, possibly due to accumulation of excess free GFP-tagged actin monomers in the array vicinity.

*5) The GFP-act88F localization around the M-bands is narrower in the sals-i, but still present. The authors should provide quantifications for the width (i.e. does this account for the difference in thin filament length) and test or at least discuss what other proteins could be responsible for the ongoing pointed end assembly in absence of sals.*

As requested, we have added quantifications in Figure 5 and Figure 5—figure supplement 1, and have examined the possible contribution of Tropomodulin to pointed end elongation (Figure 5—figure supplement 1). Please also see our answers to Reviewer #2 (points 3 and 4).